# GLOBAL MINIMA, RECOVERABILITY THRESHOLDS, AND HIGHER-ORDER STRUCTURE IN GNNS

## ABSTRACT

We analyze the performance of graph neural network (GNN) architectures from the perspective of random graph theory. Our approach promises to complement existing lenses on GNN analysis, such as combinatorial expressive power and worst-case adversarial analysis, by connecting the performance of GNNs to typical-case properties of the training data. First, we theoretically characterize the accuracy of one- and two-layer GCNs relative to the contextual stochastic block model (cSBM) and related models. We additionally prove that GCNs cannot beat linear models under certain circumstances. Second, we numerically map the recoverability thresholds, in terms of accuracy, of four diverse GNN architectures (GCN, GAT, SAGE, and Graph Transformer) under a variety of assumptions about the data. Sample results of this second analysis include: heavy-tailed degree distributions enhance GNN performance, GNNs can work well on strongly heterophilous graphs, and SAGE and Graph Transformer can perform well on arbitrarily noisy edge data, but no architecture handled sufficiently noisy feature data well. Finally, we show how both specific higher-order structures in synthetic data and the mix of empirical structures in real data have dramatic effects (usually negative) on GNN performance.

## 1 INTRODUCTION

Graph neural networks (GNNs) have achieved impressive success across many domains, including natural language processing (Wu et al., 2023a), image representation learning (Adnan et al., 2020), and perhaps most impressively in protein folding prediction (Jumper et al., 2021). GNNs' success across these fields is due to their ability to harness non-Euclidean graph topology in the learning process (Xu et al., 2019). Despite the growing use of GNN architectures, we still grapple with a significant knowledge gap concerning the intricate relationship between the statistical structure of graph data and the nuanced behavior of these models. By aligning GNN designs with data distributions, we can not only unveil the underlying mechanics and behaviors of these models but also pave the way for architectures that intuitively resonate with inherent data patterns.

While significant focus has been directed towards homophily in the context of GNN performance (Maurya et al., 2021; Halcrow et al., 2020; Zhu et al., 2020), other critical properties of graph data have remained relatively underexplored. Features such as degree distribution and mesoscale structure offer important insights into the behavior of networks. Similarly, despite the depth of theoretical advancements in graph modularity, including works such as the one by Abbe (2018), there remains a sizable gap in their integration and applicability within the GNN domain. We seek to explore such properties to bridge this gap.                                        NEW

In particular, our results imply that commonly studied properties such as homophily, Gaussian feature separation, and high dimensionality aren't enough to explain and justify the use of certain nonlinear GNNs, as we show that their performance is matched by linear GNN models in the cSBM setting. This motivates research into which significant features of the data should be incorporated into the data generation models commonly used to study GNNs.

In this paper we:                                                                    NEW

- fully characterize the accuracy of one- and two-layer GNNs satisfying certain assumptions, as well as proving that the accuracy of certain nonlinear GNNs is bounded above by the accuracy of a linear GNN when the graph is drawn from a broad family,
- report extensive numerical studies that map the degree to which edge and feature information contribute to overall performance across diverse models in a variety of random graph contexts, and
- demonstrate that the presence of higher order structures in graphs causes a dramatic (and usually negative) change in GNN accuracy.

## 2 PREVIOUS WORK

As part of this work we lay out theoretical bounds for GNN architectures. Some foundational work in our topic is as follows. Fountoulakis et al. (2023) investigated regimes in which the attention module in the Graph (GAT) (Veličković et al., 2018) makes a meaningful difference in performance. Following this, Baranwal et al. (2023) proved theoretically that using graph convolutions expands the range where a vanilla neural network can correctly classify nodes. Baranwal et al. (2021) discovered that linear classifiers on GNN embeddings generalize well to out of distribution data in stochastic block models. Lu (2022) characterized how well a GNN can separate communities on a two-class stochastic block model. Recently, Ma et al. (2022) rigorously identified noise regimes where GNNs perform well on heterophilous graphs and Chien et al. (2021) propose an architecture that adapts to the modularity of a graph. Lastly, N.T. & Maehara (2019) found that a Graph Convolutional Network (GCN) performs low pass filtering on the feature vectors and doesn't learn non-linear manifolds. [NEW]

While many have attempted to understand models through the lens of specialized data, our approach offers a unique and deeper perspective on the subject. The monograph Abbe (2018) lays out the key mathematical findings related to SBMs as they relate to community detection. Karrer & Newman (2011) developed the *degree-corrected SBM*, which allows for heavy-tailed degree distributions. Gao et al. (2018) derived asymptotic minimax risks for misclassification in degree-corrected SBMs and Mehta et al. (2019) propose a variational autoencoder for SBMs. Deshpande et al. (2018) proposed a *contextual SBM* (cSBM) that generates feature data alongside the graph data. This was originally proposed to analyze specific properties of belief propagation (Bickson, 2009). Finally, Wu et al. (2023b) not only explore the characteristics of oversmoothing in GNNs through cSBMs but also characterize how graph convolutions function both as denoising and feature-mixing mechanisms, detailing the extent and manner in which these processes occur. [NEW]

Our investigation presents a novel angle that bridges interplay between edge data and feature data. Binkiewicz et al. (2017) explored how to use features to aid spectral clustering. Yang et al. (2022) and Arroyo et al. (2021) used edges and features that contain orthogonal information to better understand the relationship between the two. While the influence of motifs or higher-order structures on GNN performance remains a hot area of exploration, our approach delves deeper into this pressing topic. Works such as Tu et al. (2020) have proposed using graphlets to aid in learning representations. Others have utilized hypergraphs to make better predictions (Huang & Yang, 2021). Much of the work quantifying the expressive power of GNNs is achieved by relating GNNs to the classical Weisfeiler-Leman (WL) heuristic for graph isomorphism (Li & Leskovec, 2022; Huang & Villar, 2021). These have inspired corresponding GNN architectures that have increased distinguishing capabilities (Hamilton, 2020)

## 3 BACKGROUND

In this work, we first theoretically determine nodewise accuracy for certain one- and two-layer GNNs and identify cases where nonlinear GNNs cannot outperform linear GNN models. We map the performance of the Graph Convolutional Network (Kipf & Welling, 2017), Graph SAGE (Hamilton et al., 2017), the Graph Attention Network (Veličković et al., 2018), and the Structure-Aware Transformer (Chen et al., 2022) on several related random graph models related to the cSBM. We will also inject and remove higher-order structure in various contexts to see how GNN performance is affected. We now describe some of the random graph models and GNN architectures on which our analysis relies. Note, when referring to data generation methods we use the term *generative models* while *model* will refer to a trained GNN. [FIX]

## 3.1 STOCHASTIC BLOCK MODELS

The stochastic block model is a random graph model that encodes node clusters ("classes") in the graph topology. The presence or absence of each edge is determined by an independent Bernoulli draw with probability determined by the class identities of the nodes. We restrict attention to SBMs where all classes have the same size and uniform inter-class and intra-class probabilities. The parameters for such an SBM are: the total number of nodes $n$, the number of equally sized classes $k$, the intra-class edge probability $p_{\text{in}}$, and the inter-class edge probability $p_{\text{out}}$. While SBMs generate realistic clustering patterns, without further modification they exhibit a binomial degree distribution. To more closely model many realistic classes of data, Karrer & Newman (2011) proposed the degree-corrected SBM, which can exhibit any degree distribution, notably heavy-tailed distributions.

In this paper, we represent edge similarity using an *edge information parameter*, $\lambda$, which has the following relationship to $p_{\text{in}}$ and $p_{\text{out}}$:

$$p_{\text{in}} = \frac{d + \lambda\sqrt{d}}{n}, \quad p_{\text{out}} = \frac{d - \lambda\sqrt{d}}{n},$$

where $d$ is the expected average node degree. Setting $\lambda = 0$ yields identical inter- and intra-class edge probabilities, meaning the topology of the graph encodes no information about class labels. A positive $\lambda$ indicates that nodes of the same class are more likely to connect than nodes of different classes (homophily), while a negative $\lambda$ indicates the reverse relationship (heterophily).

To generate node attributes, Deshpande et al. (2018) proposed the contextual SBM (cSBM), where features are drawn from Gaussian point clouds with mean at a specified distance $\mu$ from the origin. Features, $X$, are thus defined as $X(i) = \mu m_{v_i} + z_i$, where $z_i$ a standard normally distributed random variable, $v_i$ is the ground-truth class label of node $i$, and $m_{v_i}$ is the mean for class $v_i$. The means are chosen to be an orthogonal set. We can then vary the level of feature separability (feature information) by modifying $\mu$. Setting $\mu = 0$ makes node features indistinguishable across classes, while a large value of $\mu$ indicates high distinguishability. We thus refer to $\mu$ as the *feature information parameter*.

## 3.2 GRAPH NEURAL NETWORKS

As stated before, we analyze the performance of four diverse and influential architectures: GCN Kipf & Welling (2017), SAGE Hamilton et al. (2017), GAT Veličković et al. (2018), and Graph-Transformer Chen et al. (2022). In our numerical work, we also assess the performance of a standard feedforward neural network and spectral clustering (von Luxburg, 2007), which are useful points of comparison as they are agnostic to the graph and feature structures, respectively. Lastly we also use graph-tool (Peixoto, 2014) to evaluate feature-agnostic performance on heterophilous graphs.

## 4 THEORETICAL RESULTS

We now derive analytically the performance of GNN architectures when the data-generating process is known. Section 4.1 covers the one-layer case for a GCN architecture and cSBM-generated data, and section 4.2 handles the two-layer case in for a more general class of GNN architecture as well as a broader class of generating processes. We introduce the following notation first: for a given node $i$, $n_{\text{in}}(i)$ is the number of neighbors in the same class as $i$, and $n_{\text{out}}(i)$ is the number of neighbors in other classes. $\mathcal{N}(i)$ is the one-hop neighborhood of $i$. $v_i$ is the ground-truth class label of $i$. erf is the Gaussian error function. Both subsections assume a binary classification setting. The results in 4.1 are at least partly known in other literature (e.g. Lemma 1 from Wu et al. (2023b)), but they are included here for completeness.   FIX FIX NEW

## 4.1 ACCURACY ESTIMATES FOR SINGLE-LAYER GCNS

In the one-layer case, we assume the GNN is of the simple form $y[X] = \text{sign}(AXW)$, that the final embedding is into $\mathbb{R}$, and that $A$ and $X$ are generated by a cSBM, with no self-loops (but see remark 2). We also make a slight modification to the cSBM setup so that the means are diametrically

opposed rather than orthogonal. That is,

$$X(i) = \begin{cases} \mu m + z_i & \text{if node } i \text{ is in class 1} \\ -\mu m + z_i & \text{if node } i \text{ is in class 2.} \end{cases}$$

This requires no loss of generality, since all choices of two means may be translated to fit this assumption. We then have the following proposition:

**Proposition 1.** *Under the preceding assumptions, we have*

1. *For each $i$, $(AXW)_i$* has the distribution        FIX

   FIX

$$\underbrace{\mu(n_{\text{in}}(i) - n_{\text{out}}(i))m \cdot W}_{\text{neighborhood signal}} + \underbrace{\left( \sum_{j \in \mathcal{N}(i)} z_j \right) \cdot W}_{\text{noise}}.$$

2. *If $W \neq 0$, the generalization accuracy, conditioned on the graph structure is,*    FIX

$$P(y[X](i) = 1 \mid n_{\text{in}}(i), n_{\text{out}}(i), v_i = 1) = \frac{1}{2}\left( \text{erf}\left( \frac{\mu(n_{\text{in}}(i) - n_{\text{out}}(i))}{\sqrt{2(n_{\text{in}}(i) + n_{\text{out}}(i))}} \cos\theta \right) + 1 \right),$$

   *where $\theta$ is the angle between $W$ and $m$.*    FIX

3. The maximum expected accuracy for an arbitrary node *in the homophilous regime is achieved when $\theta = \pi$. In the heterophilous regime, $\theta = 0$ is the maximizer.*

*Proof.* See appendix A.    □

**Remark 1.** *Part three of this proposition shows that, in the one-layer case, optimal performance is achieved simply by aligning the learned parameters with the axis separating the means of the distributions. The proof consists largely of manipulations of the probability densities, together with calculus. A similar alignment result applies in the two-layer case, but in that case, the fastest way forward is to rely on the symmetries of the distribution and GNN, as shown below.*

**Remark 2.** *The analysis with self-loops is nearly identical, with the exception that it is possible that the maximizing parameters may possibly differ in the extremely dense, slightly heterophilous case, but this is not the regime in which GNNs are typically used.* Extremely dense refers to the case    NEW where almost all edges are present. *See the proof for full details.*

## 4.2 ANALYSIS OF TWO-LAYER GCNs    NEW

In this section, we make two claims about the effectiveness of a class of GNNs given certain symmetries in the model space. These symmetry assumptions are satisfied by the cSBM, and both results make precise that the effectiveness of nonlinear GNNs cannot be explained by cSBM-type data models. First, given a certain symmetry about the origin, we claim the cost of the model $y$ is no smaller than the cost of a linear model. Second, given an additional symmetry about any subspace $S$ of the feature space, we claim the cost of the linear model is no smaller than the cost of a projection of the linear model.

### 4.2.1 SET-UP    FIX

We define a *2-class attributed random graph model* to be a probability space $(\Omega, P)$ of tuples $(G, i, v, X)$ where $G$ is a graph, $i$ is a node in $G$, and $v$ and $X$ are functions mapping each node in the graph to its class and its feature vector, respectively. That is,

$$v : G \to \{-1, 1\}$$
$$X : G \to \mathbb{R}^{m_{\text{feat}}}$$

As a notational convenience, let $v(x)$ denote the class of the node corresponding to a tuple $x \in \Omega$.    FIX

A model $y$ on a 2-class attributed random graph model assigns to each $x \in \Omega$ a real number $y(x) \in \mathbb{R}$ that corresponds to the estimated probability that the node corresponding to $x$ is of class 1. More concretely, the predicted probability is given by

$$P(v(x) : y(x)) = \begin{cases} \sigma_s(y(x)) & v(x) = 1 \\ 1 - \sigma_s(y(x)) & v(x) = -1. \end{cases}$$

where $\sigma_s : \mathbb{R} \to (0,1)$ is is the logistic sigmoid $\sigma_s(z) = (1 + e^{-z})^{-1}$. According to maximum likelihood learning, the cost function of the model $y$ is

$$C(y) = \mathbb{E}_{x \sim \Omega}[-\log P(v(x) : y(x))].$$

In this section, we will focus on 2-layer models consisting of linear aggregators interspersed by the non-linear ReLU function $\sigma$. More concretely, a graph aggregator maps a graph and its features to a new set of features on the graph:

$$\phi : (G, X) \to X'$$

where $X' : G \to \mathbb{R}^l$ for some $l$. We write $\phi_G = \phi(G, \cdot)$. A linear aggregator (without bias) satisfies

$$\phi_G(X_1 + X_2) = \phi_G(X_1) + \phi_G(X_2)$$

for all graphs $G$ and features $X_1, X_2$. Linear aggregators include the standard sum and mean aggregators, but they also include more general aggregators such as applying the sum aggregator after adding self-loops with a custom weight. A *generalized 2-layer graph convolutional network (GCN) without bias* is then given by

$$y(x) = (\phi'_G \circ \sigma \circ \phi_G)[X](i)$$

where $\phi$ and $\phi'$ are linear aggregators, $\phi'$ maps into $\mathbb{R}$, and $\sigma$ is the ReLU function.

### 4.2.2 PRINCIPAL CLAIMS

FIX

In this section, we make two claims on the effectiveness of these generalized GCNs given certain symmetries in the model space $\Omega$. First, given a certain symmetry about the origin, we claim the cost of the model $y$ is no smaller than the cost of the linear model $L[y]$:

$$L[y](x) = \frac{1}{2}(\phi'_G \circ \phi_G)[X](i)$$

Second, given an additional symmetry about any subspace $S$ of the feature space, we claim the cost of the linear model $L[y]$ is no smaller than the cost of the projection of the linear model $P_S[L[y]]$:

$$P_S[L[y]](x) = \frac{1}{2}(\phi'_G \circ \phi_G \circ P_S)[X](i)$$

where $P_S$ is simply the projection on the subspace $S$. For example, if $\phi'$ and $\phi$ are both simply the classical right-multiplication by a weight matrix followed by summing the features of neighbors, then model $y$ becomes

$$y(x) = \sum_{j \in \mathcal{N}(x)} \sigma \left( \sum_{k \in \mathcal{N}(j)} X(k)W \right) \cdot c$$

for some weight matrix $W$ and weight vector $c$. If the symmetries mentioned above hold for the subspace $S = \mathrm{span}\{\vec{m}\}$ for some vector $\vec{m}$ (as is the case with a cSBM), then the above claims assert the cost of the model $y$ is no smaller than the cost of the model,

$$P_S[L[y]](x) = \frac{1}{2} \sum_{j \in \mathcal{N}(x)} \sum_{k \in \mathcal{N}(j)} P_S(X(k))W \cdot c = K \sum_{j \in \mathcal{N}(x)} \sum_{k \in \mathcal{N}(j)} X(k) \cdot \vec{m}$$

for some $K \in \mathbb{R}$.

FIX

The first symmetry is defined using the negation of element of $\Omega$. If $x = (G, i, v, X) \in \Omega$, we define the negation of $x$ to be the tuple $-x = (G, i, -v, -X)$. In other words, $x$ has the same graph with all of the classes and features negated. We similarly define the negation of a subset $F \subset \Omega$ by $-F = \{-x : x \in F\}$. We say a 2-class attributed random graph model is *class-symmetric about the origin* if $P(F) = P(-F)$ for all measurable $F \subset \Omega$. Heuristically, this means that in the distribution of graphs, the nodes of the two classes have the same topological distribution (which still allows for homophily/heterophily) and that the feature distribution of of class -1 is equal to the feature distribution of class 1 reflected across the origin. A cSBM with an equal number of nodes in both classes satisfies

this symmetry, but this property is also held by graph models having non-Gaussian noise so long as there is symmetry across the origin.

FIX

The second symmetry concerns the feature distribution alone. If $S$ is subspace of the feature space and $R_S$ is the reflection across $S$, then the reflection of $x \in \Omega$ is defined by $R_S(x) = (G, i, v, R_S \circ X)$. In other words, $x$ has the same graph with all the features reflected across $S$. We similarly define the reflection of a subset $F \subset \Omega$ by $R_S(F) = \{R_S(x) : x \in F\}$. We say a 2-class attributed random graph model is *symmetric about* $S$ if $P(F) = P(R_S(F))$ for all measurable $F \subset \Omega$. Heuristically, this means the feature distribution is symmetric about the subspace $S$.

**Theorem 1.** *Let $\Omega$ be a 2-class attributed random graph model and let $y$ be any two-layer generalized GCN without bias on $\Omega$. If $\Omega$ is class-symmetric about the origin then,*

$$C(L[y]) \leq C[y].$$

*Furthermore, if $\Omega$ is symmetric about $S$ then,*

$$C(P_S[L[y]]) \leq C(L[y])$$

*Proof.* See appendix B. The main idea is to use the symmetries of the space together with the convexity of the objective to invoke Jensen's inequality. $\square$

NEW

We note that similarity between the previous theorem and ideas from Wu et al. (2023b). Our work focuses on models with a stacked non-linearity, while the latter deals primarily with linear models.

In light of the the preceding theorem, linear GCNs are optimal over the binary cSBM. Carefully analyzing the linear case, we obtain an explicit formula for the optimal accuracy of any GCN over cSBM data. Although difficult to analyze theoretically, the accuracy can calculated empirically using the following formula (see the remark afterward for an intuitive explanation):

NEW

**Theorem 2.** *In the large node limit of a cSBM, the linear model*

$$y(x) = K \sum_{j \in \mathcal{N}(x)} \sum_{k \in \mathcal{N}(j)} X(k) \cdot m$$

*has accuracy*

$$\sum_{n_{\mathrm{in}}, n_{\mathrm{out}}, n_{2-\mathrm{in}}, n_{2-\mathrm{out}}=0}^{\infty} P(n_{\mathrm{in}}, n_{\mathrm{out}}, n_{2-\mathrm{in}}, n_{2-\mathrm{out}}) \Phi\left(\psi\left(\mathrm{sgn}(K)\tfrac{\mu}{\sigma}, n_{\mathrm{in}}, n_{\mathrm{out}}, n_{2-\mathrm{in}}, n_{2-\mathrm{out}}\right)\right)$$

*where $\Phi$ is the cdf of the standard normal distribution and the following definitions apply:*

$$P(n_{\mathrm{in}}, n_{\mathrm{out}}, n_{2-\mathrm{in}}, n_{2-\mathrm{out}})$$
$$= p(n_{\mathrm{in}}, d_{\mathrm{in}}) \cdot p(n_{\mathrm{out}}, d_{\mathrm{out}}) \cdot p(n_{2-\mathrm{in}}, d_{\mathrm{in}}n_{\mathrm{in}} + d_{\mathrm{out}}n_{\mathrm{out}}) \cdot p(n_{2-\mathrm{out}}, d_{\mathrm{out}}n_{\mathrm{in}} + d_{\mathrm{in}}n_{\mathrm{out}}),$$
$$p(k, \lambda) = \frac{\lambda^k e^{-\lambda}}{k!}, \ and$$
$$\psi(c, n_{\mathrm{in}}, n_{\mathrm{out}}, n_{2-\mathrm{in}}, n_{2-\mathrm{out}}) = c\frac{1 + 3n_{\mathrm{in}} - n_{\mathrm{out}} + n_{2-\mathrm{in}} - n_{2-\mathrm{out}}}{\sqrt{(n_{\mathrm{in}} + n_{\mathrm{out}} + 1)^2 + 4(n_{\mathrm{in}} + n_{\mathrm{out}}) + (n_{2-\mathrm{in}} + n_{2-\mathrm{out}})}}.$$

*Proof.* See appendix B. $\square$

**Remark 3.** *In the theorem, the indices $n_{\mathrm{in}}$, $n_{\mathrm{out}}$, $n_{2-\mathrm{in}}$, and $n_{2-\mathrm{out}}$ refer to the number of distance 1 and 2 nodes with the same and the opposite class of the base node.* The function $P$ represents the probability of the graph structure having such characteristics, while the function $\Phi \circ \psi$ is the accuracy at the base node given such characteristics. The function $p$ is the p.m.f. of the Poisson distribution.

FIX

## 5 EMPIRICAL EXPLORATION OF DATA REGIMES

In section 5.1 and section 5.2, we present results from our simplest set of experiments in detail to illustrate the interplay between edges and features. Then, in section 5.3 we compare performance across each of the four architectures. Finally, we contrast how GNNs performed on degree-corrected and non-degree-corrected graphs in section 5.4. See also our full code online to extend this work to other architectures and parameter ranges:

## 5.1 Experimental design

To better understand how GNN architectures harness information embedded in the features or edges, we evaluated them across a variety of graphs. Each of our architectures was comprised of one input layer, a hidden layer of size 16 (with ReLU activation functions), and an output layer (with softmax). As baselines, we trained a feedforward neural network, with one hidden layer of size 16, on the feature data. Our exploration also encompassed a variety of methods for feature-agnostic methods such as graph-tool (Peixoto, 2014), Leidenalg (python package), Louvian (python package), and Spectral clustering (Pedregosa et al., 2011). In doing so we found that spectral clustering worked the best for assortative graphs (edge information from [0,3]) and graphtool performed the best on dissasortative graphs (edge information from [-3,0)).

We generated graph data using a cSBM with average degree $d = 10$; the number of nodes $n = 1,000$; the number of features $m_{\text{feat}} = 10$; the number of classes $c = 2$; and standard deviation of the Gaussian clouds .2. These hyperparameters were selected to be representative of a large variety of datasets without being too computationally expensive (specifically when using transformers). We observed that $1,000$ nodes was large enough to get statistical regularity and that using larger graphs (up to 40,000 nodes) didn't introduce major deviations. With these hyperparameters, we vary $\lambda$ (edge separation in cSBMs) between $-3$ and $3$ and vary feature separation (cloud distance from origin) from 0 to 2 to obtain $121 \times 200$ (how finely we discretized the interval) possible sets of graph data. This data ranges from being highly disassortative to highly assortative.

To train each architecture, we used an Adam optimizer (PyTorch) with a learning rate of $0.01$ for $400$ epochs (typically where the model ceased improving). We evaluated the final accuracy on a separate graph, with the same graph parameters to prevent overfitting.

In addition to the class count of two, we ran the architectures across class counts of three, five, and seven each with both a degree-corrected case and a binomial case. As each test was averaged/maxed over 10 trials, the number of tests totals 320 different tests with $15,488,000$ accuracy scores generated (more than .25 petaflops used in total). We note that we used two hidden layers and Gaussian     NEW
distributions for simplicity, but more complex distributions and additional layers merit future research.

## 5.2 Example: Binary node classification with graph transformer

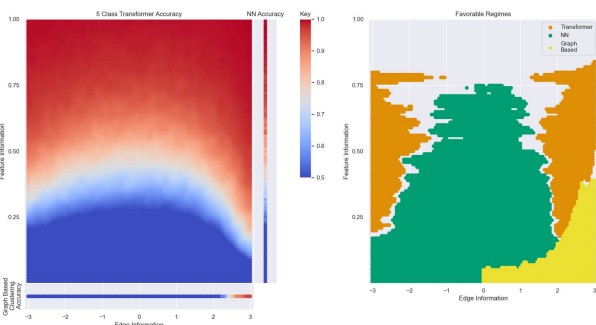

Figure 1: (Left) Transformer's performance on a five-class non-degree-corrected cSBM, with color gradients indicating accuracy levels. To the right and below, performance curves for the feedforward neural network (graph-blind) and graph-based (feature-blind) methodologies are displayed respectively. (Right) A comparison of the top-performing model among the Graph Transformer, feedforward neural network, and graph-based clustering. White space indicates where one model was not consistently better than the others. The Transformer predominantly excels when edge and feature information were moderately noisy. The graph based method is able to surpass the transformer if we have a combination of high feature noise and low edge noise.

Our experiments with the Transformer architecture elucidate its robustness across a wide parameter space (see fig. 1). Remarkably, the Transformer consistently delivers superior performance across

most scenarios, with exceptions only in cases where both the feature and edge information are heavily compromised by noise. An intriguing capability of the Transformer is its potential to achieve flawless accuracy even when presented with solely noisy edge information. This implies an innate adaptability within the Transformer to sift through the noise, selectively emphasizing pertinent features over less informative edges. Message-passing GNNs seem to struggle with this (Bechler-Speicher et al., 2023) as seen in fig. 2.

The Transformer performs well on heterophilous graphs as well, most clearly seen in fig. 2. Such proficiency makes the Transformer an excellent candidate for tasks demanding the assimilation of diverse or opposing sets of information. A marked limitation is observed in the Transformer's ability to process noisy feature scenarios, where spectral clustering performs better. The Transformer's somewhat dependent relationship with feature information, even when suboptimal, necessitates further investigation.                                    NEW

### 5.3   PERFORMANCE OF GCN, GAT, SAGE, AND TRANSFORMER ARCHITECTURES

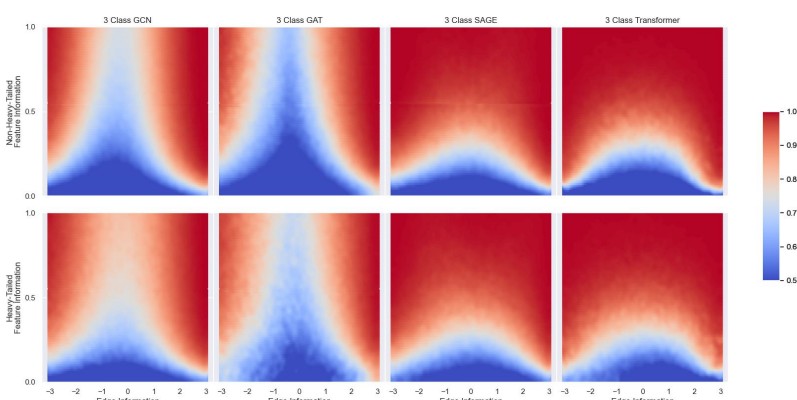

Figure 2: Comparison performance on non-degree-corrected and degree-corrected SBMs for GCN, GAT, SAGE and Transformer architectures. Notice the GCN and GAT consistently perform worse when the edge information is roughly zero, but the other two models achieve perfect accuracy given enough feature information. This could be due to SAGE and Transformer learning a more global context for each node. In this regime we see that almost all of the models did better on the heavy tailed graphs. GCN achieved higher accuracy on such graphs when the edges were just noise. The accuracy of the GAT improved as well in the regime of very noisy edges and features. All values are the best of 10 trials, with a $5 \times 5$ convolutional filter applied for visual clarity.

We now juxtapose the performances of four distinct architectures, particularly considering the influence of heavy-tailed degree distributions. Refer to fig. 2 for insights on the three-class scenario, while an exhaustive analysis is cataloged in appendix C.1 and appendix C.2. Generally, both Graph-Transformer and SAGE stand out for their resistance to edge and feature noise, demonstrating their robustness in noisy regimes. In a three-class, non-degree-corrected cSBM setting, SAGE and Graph-Transformer consistently outperform the other two models, GAT and GCN. This is shown by their strong resistance to feature noise and their ability to classify accurately even without edge information. Such performance highlights SAGE's use of global information from random walks and graph embeddings, while the Transformer simply ignores the graph embedding.                    FIX / NEW

Each architecture performs differently, as shown by their varying weak areas (seen as blue areas in fig. 2) and how they compare to neural network and spectral clustering benchmarks (detailed in appendix C.2). The GAT and GCNs weak area is especially prominent with no edge information, showing it relies heavily on clear features. Interestingly, both Transformer and GAT perform better with degree correction, especially in heterophilous settings. For a more in depth comparison of different models see appendix C.2                    NEW

## 5.4 DEGREE-CORRECTED SBMs

We found that all models performed better on scale-free graphs. We believe this occurs due to a filtering out of bad neighbors. Most nodes in the heavy-tailed data have relatively few neighbors, this allows for fewer confusing neighbors to contribute misleading information in the aggregation step than in the binomial degree distribution. This is similar to ideas from Albert et al. (2000).

The scale free graphs affected the models in different ways, for example the performance of SAGE only improved in the higher signal edge regimes (right and left sides of the fig. 2). The performance of GAT increased dramatically in the case of very noisy edges and features. This is likely because degree correction gave it more information on what edges to prune. Interestingly, the attention based models, the Transformer and GAT, saw a stark increase in performance in the heterophilous clustering, suggesting that self-attention allows for a better interpretation of such graphs.

## 6 EFFECT OF HIGHER-ORDER STRUCTURE IN REAL WORLD DATASETS

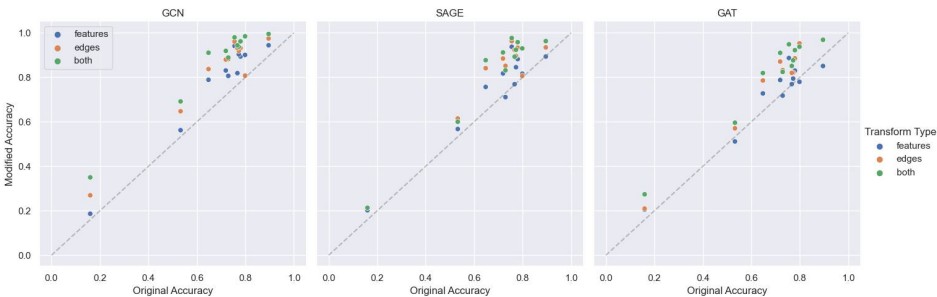

Figure 3: Comparison of model accuracies on real data compared to performance on matched synthetic data. The accuracy tends to improve when we erase higher-order structure in the data. The datasets from left to right are: Flickr, DeezerEurope, Citeseer, LastFMAsia, DBLP, Facebook-PagePage, Pubmed, GitHub, Cora, Amazon Computers, and Amazon Photos. The figure depicts cases where we transform only the edges, only the features, and both. The transformer was not run due to memory requirements.

The experiments to be described in this section support the claim that higher-order structure, such as clustering or motifs, influence the performance of GNN architectures. We found that the models generally performed better on matched synthetic data than on real data, suggesting that the higher-order structure that was erased is an impediment to GNN learning (see fig. 3).

To make the synthetic data for each data set, we transformed the edge and feature data as if each dataset were already a degree-corrected cSBM. we used a variety of datasets from pytorch geometric (Fey & Lenssen, 2019). In particular, the edge data was randomized by rewiring every edge to preserve degree distribution and modularity similar to ideas in Fosdick et al. (2018). In some experiments, the node features were also transformed by sampling from the estimated normal distribution. Thus, the synthetic data lacks nontrivial structure except the structure implied by the degree distribution, intra/inter-class linkage frequency, and feature means and standard deviations match the corresponding empirical network.     FIX  NEW  FIX

We see a positive impact on the accuracy of the GCN when removing the higher-order structure (see fig. 3) specifically with edge structure. The fact that the GNNs do better on this semi-randomized data suggests that they may perform optimally on SBM-like data, but are negatively impacted by the additional structure present in real data. Uncovering why such structure can be detrimental to these GNNs is a significant opportunity for future work.

To further verify that we are not confusing higher-order structure with label noise, we verified these results on synthetic data with controlled structure. Such results indicate that GNNs perform worse on datasets with spatial structure, but are unaffected by local motifs such as triadic closure. Results on graphs with planted hierarchical structure were mixed but largely favored SBM data. A more detailed analysis can be found in appendix D.

## 7 REPRODUCIBILITY STATEMENT

For further explanation of various proofs explored in section 4, see appendix A and appendix B. For code implementations of our studies in section 5.3 and section 6, see our GitHub or the supplementary material. For the exact implementation of section 5.3, view the hyperparameters discussed in section 5.1. In regards to our findings in section 6, view appendix D for a more in-depth explanation.

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

# A   ANALYSIS OF ONE-LAYER GCN

In this appendix, we prove the three parts of proposition 1.

## A.1   DISTRIBUTION OF LINEAR EMBEDDINGS

Analyzing the linear part of the model gives

$$(AXW)_i = \sum_{j \in \mathcal{N}(i)} X(j)W.$$

From here, we split the sum into two parts corresponding to the two possible classes of neighbors:

$$\sum_{\substack{j \in \mathcal{N}(i) \\ j \text{ in class } 1}} X(j)W + \sum_{\substack{j \in \mathcal{N}(i) \\ j \text{ in class } 2}} X(j)W$$

We then substitute the known expressions for $X(j)$:

$$\sum_{\substack{j \in \mathcal{N}(i) \\ j \text{ in class } 1}} \left(\mu m + z_j\right)W + \sum_{\substack{j \in \mathcal{N}(i) \\ j \text{ in class } 2}} \left(-\mu m + z_j\right)W.$$

This becomes

$$(AXW)_i = \underbrace{\mu(n_{\text{in}} - n_{\text{out}})mW}_{\text{neighborhood signal}} + \underbrace{\left(\sum_{j \in \mathcal{N}(i)} z_j\right)W}_{\text{noise}}.$$

## A.2   NODEWISE ACCURACY, CONDITIONED ON THE GRAPH STRUCTURE

Assume $W \neq 0$. If $n_{\text{in}} = n_{\text{out}} = 0$ then we have an isolated point. Since we are assuming no self loops and have no bias, these nodes do not affect the optimal parameters (in particular, the convolution outputs zero for these nodes). Thus we can assume that each node has at least one edge. We then compute,

$$P(y[X](i) = 1 \mid n_{\text{in}}, n_{\text{out}}, v_i = 1) = \int_0^\infty \frac{1}{\sqrt{2\pi W^T W(n_{\text{in}} + n_{\text{out}})}} e^{-\frac{1}{2}\left(\frac{x - (\mu(n_{\text{in}} - n_{\text{out}})mW)}{\sqrt{W^T W(n_{\text{in}} + n_{\text{out}})}}\right)^2} dx \tag{1}$$

We now fix,

$$u = \frac{x - (\mu(n_{\text{in}} - n_{\text{out}})mW)}{\sqrt{2W^TW(n_{\text{in}} + n_{\text{out}})}} \text{ with } du = \frac{dx}{\sqrt{2W^TW(n_{\text{in}} + n_{\text{out}})}}. \tag{2}$$

Notice $\sqrt{2W^TW(n_{\text{in}} + n_{\text{out}})} > 0$ since each node has at least one edge and $W \neq 0$. We have,

$$\frac{\sqrt{2W^TW(n_{\text{in}} + n_{\text{out}})}}{\sqrt{2\pi W^TW(n_{\text{in}} + n_{\text{out}})}} \int_{\frac{-(\mu(n_{\text{in}} - n_{\text{out}})mW)}{\sqrt{2W^TW(n_{\text{in}}+n_{\text{out}})}}}^{\infty} e^{-u^2} du$$

$$= \frac{1}{2} \int_{\frac{-(\mu(n_{\text{in}} - n_{\text{out}})mW)}{\sqrt{2W^TW(n_{\text{in}}+n_{\text{out}})}}}^{\infty} \frac{2e^{-u^2}}{\sqrt{\pi}} du = \frac{1}{2}\text{erf}(u)\Big|_{\frac{-(\mu(n_{\text{in}} - n_{\text{out}})mW)}{\sqrt{2W^TW(n_{\text{in}}+n_{\text{out}})}}}^{\infty}$$

Observe that $\lim_{u \to \infty} \text{erf}(u) = 1$, so

$$\frac{1}{2}\left(\lim_{u \to \infty} \text{erf}(u) - \text{erf}\left(\frac{-(\mu(n_{\text{in}} - n_{\text{out}})mW)}{\sqrt{2W^TW(n_{\text{in}} + n_{\text{out}})}}\right)\right)$$

$$= \frac{1}{2}\left(\text{erf}\left(\frac{\mu(n_{\text{in}} - n_{\text{out}})mW}{\sqrt{2W^TW(n_{\text{in}} + n_{\text{out}})}}\right) + 1\right).$$

Thus we have that

$$P(y[X](i) = 1 \mid n_{\text{in}}, n_{\text{out}}, v_i = 1) = \frac{1}{2}\left(\text{erf}\left(\frac{\mu(n_{\text{in}} - n_{\text{out}})mW}{\sqrt{2W^TW(n_{\text{in}} + n_{\text{out}})}}\right) + 1\right),$$

as promised.

### A.3 Maximizing accuracy

Given the symmetry of the linear model,

$$P(y_i = v_i) = P(y_i = v_i|v_i = 1)$$

Let $P_{\text{in}}(n_{\text{in}})$ be the probability of having $n_{\text{in}}$ homophilous edges, and $P_{\text{out}}(n_{\text{out}})$ be the probability of having $n_{\text{out}}$ heterophilous edges. Since $P_{\text{in}}(n_{\text{in}})$ and $P_{\text{out}}(n_{\text{out}})$ are independent we have,

$$P(y_i = v_i|v_i = 1) = P((AXW)_i > 0 \mid v_i = 1) \tag{3}$$

$$= \sum_{n_{\text{in}}=0}^{\frac{N}{2}} \sum_{n_{\text{out}}=0}^{\frac{N}{2}} P((AXW)_i > 0 \mid n_{\text{in}}, n_{\text{out}})P_{\text{in}}(n_{\text{in}})P_{\text{out}}(n_{\text{out}}). \tag{4}$$

Recall that $\theta$ is the angle between $W$ and $m$ and that consequently $\cos\theta = mW/\sqrt{W^TW}$. To find the maximizers, we now differentiate each term $P((AXW)_i > 0 \mid n_{\text{in}}, n_{\text{out}})$ with respect to $\theta$ and set it equal to 0.

$$\frac{d}{d\theta}\left(\frac{1}{2}\left(\text{erf}\left(\frac{\mu(n_{\text{in}} - n_{\text{out}})}{\sqrt{2(n_{\text{in}} + n_{\text{out}})}}\cos(\theta)\right) + 1\right)\right) = 0 \tag{5}$$

$$-\frac{\mu(n_{\text{in}} - n_{\text{out}})}{\sqrt{2(n_{\text{in}} + n_{\text{out}})}}\frac{1}{2}\sin(\theta)\left(\text{erf}'\left(\frac{\mu(n_{\text{in}} - n_{\text{out}})}{\sqrt{2(n_{\text{in}} + n_{\text{out}})}}\cos(\theta)\right)\right) = 0, \tag{6}$$

$$-\frac{\mu(n_{\text{in}} - n_{\text{out}})}{\sqrt{2(n_{\text{in}} + n_{\text{out}})}}\frac{1}{2}\sin(\theta)\frac{2}{\sqrt{\pi}}e^{-\left(\frac{\mu(n_{\text{in}}-n_{\text{out}})}{\sqrt{2(n_{\text{in}}+n_{\text{out}})}}\cos(\theta)\right)^2} = 0 \tag{7}$$

where $\text{erf}'(x) = \frac{2}{\sqrt{\pi}}e^{-x^2}$. This is equal to 0 exactly when $\theta = 0$ and $\theta = \pi$.

It turns out these are the only two critical values. To demonstrate this we reintroduce our summations and re-index $\alpha = n_{\text{in}} + n_{\text{out}}$ and $\beta = n_{\text{in}} - n_{\text{out}}$. Define $U(\alpha) = \frac{N}{2} - |\alpha - \frac{N}{2}|$:

$$\sum_{\alpha=0}^{N} \sum_{\beta=-U(\alpha)}^{U(\alpha)} -\frac{\mu\beta}{\sqrt{2\alpha\pi}} \sin(\theta) e^{-\frac{\mu^2\beta^2}{2\alpha}\cos^2(\theta)} P_{\text{in}}(n_{\text{in}}(\alpha,\beta)) P_{\text{out}}(n_{\text{out}}(\alpha,\beta)) \tag{8}$$

$$= \frac{-\mu\sin(\theta)}{\sqrt{2\pi}} \sum_{\alpha=0}^{N} \sum_{\beta=-U(\alpha)}^{U(\alpha)} \frac{\beta}{\sqrt{\alpha}} e^{-\frac{\mu^2\beta^2}{2\alpha}\cos^2(\theta)} P_{\text{in}}(n_{\text{in}}(\alpha,\beta)) P_{\text{out}}(n_{\text{out}}(\alpha,\beta)) \tag{9}$$

by pairing off entries whose absolute value of beta are equal we have:

$$= \frac{-\mu\sin(\theta)}{\sqrt{2\pi}} \sum_{\alpha=0}^{N} \sum_{\beta=1}^{U(\alpha)} c(\alpha,\beta,\theta) \left( P_{\text{in}}(n_{\text{in}}(\alpha,\beta)) P_{\text{out}}(n_{\text{out}}(\alpha,\beta)) - P_{\text{in}}(n_{\text{out}}(\alpha,\beta)) P_{\text{out}}(n_{\text{in}}(\alpha,\beta)) \right) \tag{10}$$

with $c(\alpha,\beta,\theta) = \frac{\beta}{\sqrt{\alpha}} e^{-\frac{\mu^2\beta^2}{2\alpha}\cos^2(\theta)}$. From here note that in the case of homophily:

$$P_{\text{in}}(n_{\text{in}}(\alpha,\beta)) P_{\text{out}}(n_{\text{out}}(\alpha,\beta)) - P_{\text{in}}(n_{\text{out}}(\alpha,\beta)) P_{\text{out}}(n_{\text{in}}(\alpha,\beta)) > 0 \tag{11}$$

and heterophily:

$$P_{\text{in}}(n_{\text{in}}(\alpha,\beta)) P_{\text{out}}(n_{\text{out}}(\alpha,\beta)) - P_{\text{in}}(n_{\text{out}}(\alpha,\beta)) P_{\text{out}}(n_{\text{in}}(\alpha,\beta)) < 0. \tag{12}$$

To see this, note that

$$P_{\text{in}}(n_{\text{in}}) P_{\text{out}}(n_{\text{out}}) \propto \binom{\frac{N}{2}}{n_{\text{in}}} \binom{\frac{N}{2}}{n_{\text{out}}} p_{\text{in}}^{n_{\text{in}}} p_{\text{out}}^{n_{\text{out}}}.$$

Similarly,

$$P_{\text{in}}(n_{\text{out}}) P_{\text{out}}(n_{\text{in}}) \propto \binom{\frac{N}{2}}{n_{\text{in}}} \binom{\frac{N}{2}}{n_{\text{out}}} p_{\text{in}}^{n_{\text{out}}} p_{\text{out}}^{n_{\text{in}}}.$$

Subtracting yields

$$P_{\text{in}}(n_{\text{in}}) P_{\text{out}}(n_{\text{out}}) - P_{\text{in}}(n_{\text{out}}) P_{\text{out}}(n_{\text{in}}) \tag{13}$$

$$\propto \binom{\frac{N}{2}}{n_{\text{in}}} \binom{\frac{N}{2}}{n_{\text{out}}} (p_{\text{in}}^{n_{\text{in}}} p_{\text{out}}^{n_{\text{out}}} - p_{\text{in}}^{n_{\text{out}}} p_{\text{out}}^{n_{\text{in}}}) \tag{14}$$

$$= \binom{\frac{N}{2}}{n_{\text{in}}} \binom{\frac{N}{2}}{n_{\text{out}}} p_{\text{in}}^{n_{\text{out}}} p_{\text{out}}^{n_{\text{in}}} \left( \left(\frac{p_{\text{in}}}{p_{\text{out}}}\right)^{n_{\text{in}} - n_{\text{out}}} - 1 \right). \tag{15}$$

Since $n_{\text{in}} \geq n_{\text{out}}$ (because $\beta \geq 0$), eq. (15) is positive in the heterophilous case and negative otherwise (unless $n_{\text{in}} = n_{\text{out}}$, of course). If $n_{\text{in}} = n_{\text{out}}$ then the probabilty for any node to be classified correctly is .5 regardless of the weight matrix, so every weight matrix is optimal. NEW

In any case, the first derivative is not equal to zero unless $\theta \in \{0, \pi\}$.

Notice if we include self-loops, the analysis case is very similar, with the caveat that there may rarely be another critical point in the very dense heterophilous case, due to the possibility of $n_{\text{in}} = \frac{N}{2}$.

Thus the critical points are $0$ and $\pi$.

We now take the second derivative with respect to $\theta$ to classify the critical points. For clarity we set $h = \frac{\mu(n_{\text{in}} - n_{\text{out}})}{\sqrt{2(n_{\text{in}} + n_{\text{out}})}}$. Again, proceeding term by term gives

$$\frac{-h}{\sqrt{\pi}} \frac{d}{d\theta} \left( \sin(\theta) e^{-h^2\cos^2\theta} \right) \tag{16}$$

$$= \frac{-2h^3}{\sqrt{\pi}} \sin(\theta)^2 \cos(\theta) e^{-h^2\cos^2\theta} - \frac{h}{\sqrt{\pi}} \cos(\theta) e^{-h^2\cos^2\theta}. \tag{17}$$

Since $\sin(\theta) = 0$ at both critical points and $\cos(\theta) = \pm 1$, this simplifies to

$$= -\frac{h}{\sqrt{\pi}}\cos(\theta)e^{-h^2}. \tag{18}$$

We now reintroduce the summations and reindex. Once again fixing $\alpha = n_{\text{in}} + n_{\text{out}}$ and $\beta = n_{\text{in}} - n_{\text{out}}$. Let $U(\alpha)$ be defined as above. The second derivative is then

$$\frac{-\mu}{\sqrt{2\pi}}\cos(\theta)\sum_{\alpha=0}^{N}\sum_{\beta=-U(\alpha)}^{U(\alpha)}\frac{\beta}{\sqrt{\alpha}}e^{-\frac{\mu^2\beta^2}{2\alpha}}P(n_{\text{in}}(\alpha,\beta))P(n_{\text{out}}(\alpha,\beta)) \tag{19}$$

$$= \frac{-\mu}{\sqrt{2\pi}}\cos(\theta)\sum_{\alpha=0}^{N}\sum_{\beta=1}^{U(\alpha)}\frac{\beta}{\sqrt{\alpha}}e^{-\frac{\mu^2\beta^2}{2\alpha}}\left(P_{\text{in}}(n_{\text{in}}(\alpha,\beta))P_{\text{out}}(n_{\text{out}}(\alpha,\beta)) - P_{\text{in}}(n_{\text{out}}(\alpha,\beta))P_{\text{out}}(n_{\text{in}}(\alpha,\beta))\right) \tag{20}$$

Similar to the analysis with the first derivative, the second term in the innermost sum is always less than the first (assuming homophily here), we that the second derivative must be positive at $\pi$ and negative at $0$, as expected. In the heterophilous case, the opposite sign rules apply.

Thus in the homophilous case the maximal accuracy is obtained when $\theta = 0$ or our weight matrix is pointing in the same direction as our average feature vector. The minimal accuracy is obtained with $\theta = \pi$. For heterophily reversed rules apply.

# B  ANALYSIS OF TWO-LAYER GNNS

## B.1  PROOF THAT LINEAR MODELS ARE OPTIMAL IN CERTAIN CASES

Let $\Omega$ be a 2-class attributed random graph model. For any $x = (G, i, v, X) \in \Omega$, we defined earlier the negation
$$-x = (G, i, -v, -X)$$
and the reflection
$$R_S(x) = (G, i, v, R_S \circ X)$$
where $S$ is some subspace of $\mathbb{R}^m$. We also define $P_\perp = I - P_S$ or the projection onto the subspace orthogonal to $S$.

**Lemma 1.** *For any model $y$ on $\Omega$, the cost function is given by:*

$$C(y) = \mathbb{E}_{x\sim\Omega}\left[\log\left(1 + e^{-v(x)y(x)}\right)\right]$$

*Proof.* By definition,
$$C(y) = \mathbb{E}_{x\sim\Omega}[-\log P(v(x) : y(x))]$$
where
$$P(v(x) : y(x)) = \begin{cases} \sigma_s(y(x)) & v(x) = 1 \\ 1 - \sigma_s(y(x)) & v(x) = -1 \end{cases}$$

and $\sigma_s(z) = (1 + e^{-z})^{-1}$. Famously the sigmoid function satisfies $1 - \sigma_s(z) = \sigma_s(-z)$. We can then re-write the probability as

$$P(v(x) : y(x)) = \sigma_s(v(x)y(x))$$

Using the additionally identity $-\log\sigma_s(z) = \log(1 + e^{-z})$ we obtain,

$$C(y) = \mathbb{E}_{x\sim\Omega}\left[-\log\sigma_s(v(x)y(x))\right] = \mathbb{E}_{x\sim\Omega}\left[\log\left(1 + e^{-v(x)y(x)}\right)\right]$$

$\square$

**Lemma 2.** *The function $f(x) = \log(1 + e^{-x})$ is convex.*

*Proof.* It suffices to take the second derivative:

$$f''(x) = \frac{e^x}{(1 + e^x)^2} > 0.$$

$\square$

**Lemma 3.** *Let $y$ be any model on $\Omega$. If $\Omega$ is class-symmetric about the origin, then following inequality holds:*

$$C(y) \geq \mathbb{E}_{x \sim \Omega} \left[ \log \left( 1 + e^{-v(x)\frac{y(x) - y(-x)}{2}} \right) \right]$$

*If $\Omega$ is symmetric about the subspace $S$, then*

$$C(y) \geq \mathbb{E}_{x \sim \Omega} \left[ \log \left( 1 + e^{-v(x)\frac{y(x) + y(R_S(x))}{2}} \right) \right]$$

*Proof.* First let $\Omega$ be class-symmetric about the origin. By the above lemma,

$$C(y) = \mathbb{E}_{x \sim \Omega} \left[ \log \left( 1 + e^{-v(x)y(x)} \right) \right]$$

Since $P(F) = P(-F)$ for all $F \subset \Omega$, we can make a change of variables $x \mapsto -x$ to obtain

$$C(y) = \mathbb{E}_{x \sim \Omega} \left[ \log \left( 1 + e^{-v(-x)y(-x)} \right) \right] = \mathbb{E}_{x \sim \Omega} \left[ \log \left( 1 + e^{v(x)y(-x)} \right) \right]$$

We may therefore add the two expressions and divide by 2 to arrive at,

$$C(y) = \frac{1}{2} \mathbb{E}_{x \sim \Omega} \left[ \log \left( 1 + e^{-v(x)y(x)} \right) + \log \left( 1 + e^{v(x)y(-x)} \right) \right].$$

By Jensen's inequality for convex functions,

$$\frac{1}{2} \left[ f(z_1) + f(z_2) \right] \geq f \left( \frac{z_1 + z_2}{2} \right)$$

for $f(z) = \log(1 + e^{-z})$. Applying this to $C[y]$, we obtain

$$C(y) \geq \log \left( 1 + e^{\frac{1}{2}(-v(x)y(x) + v(x)y(-x))} \right) \tag{21}$$

$$= \log \left( 1 + e^{-v(x)\frac{y(x) - y(-x)}{2}} \right) \tag{22}$$

If $\Omega$ is symmetric about the subspace $S$, then the same reasoning yields,

$$C(y) \geq \log \left( 1 + e^{-v(x)\frac{y(x) + y(R_S(x))}{2}} \right).$$

Note that there is a sign difference from the previous expression, as negation flips the classes while reflection does not. $\square$

Recall that if

$$y(x) = (\phi'_G \circ \sigma \circ \phi_G)[X](i)$$

then

$$L[y](x) = \frac{1}{2}(\phi'_G \circ \phi_G)[X](i)$$

and

$$P_S[L[y]](x) = \frac{1}{2}(\phi'_G \circ \phi_G \circ P_S)[X](i)$$

**Lemma 4.** *Let $y$ be any generalized 2-layer GCN without bias on $\Omega$. Then for any $x \in \Omega$,*

$$\frac{y(x) - y(-x)}{2} = L[y](x)$$

*and for any subspace $S$ of $\mathbb{R}^m$,*

$$\frac{L[y](x) + L[y](R_S(x))}{2} = R_S[L[y]](x)$$

*Proof.*

$$y(x) - y(-x) = (\phi'_G \circ \sigma \circ \phi_G)[X](i) - (\phi'_G \circ \sigma \circ \phi_G)[-X](i) \tag{23}$$

$$= \phi'_G\big(\sigma(\phi_G(X)) - \sigma(\phi_G(-X))\big)(i) \quad \text{(by linearity of } \phi'_G) \tag{24}$$

$$= \phi'_G\big(\sigma(\phi_G(X)) - \sigma(-\phi_G(X))\big)(i) \quad \text{(by linearity of } \phi_G) \tag{25}$$

$$= \phi'_G(\phi_G(X))(i) \quad \text{(as } \sigma(z) - \sigma(-z) = z) \tag{26}$$

$$= 2L[y](x) \tag{27}$$

which after dividing by 2 proves the first expression. Next,

$$2(L[y](x) + L[y](R_S(x))) \tag{28}$$

$$= (\phi'_G \circ \phi_G)[X](i) + (\phi'_G \circ \phi_G)[R_S(X)](i) \tag{29}$$

$$= (\phi'_G \circ \phi_G)[P_S(X) + P_\perp(X)](i) + (\phi'_G \circ \phi_G)[P_S(X) - P_\perp(X)](i) \tag{30}$$

$$= 2(\phi'_G \circ \phi_G)[P_S(X)](i) \quad \text{(by linearity of } \phi'_G \text{ and } \phi_G) \tag{31}$$

$$= 4P_S[L[y]](x) \tag{32}$$

which after dividing by 4 proves the second expression. $\square$

**Theorem 3.** *Let $\Omega$ be a 2-class attributed random graph model and let $y$ be any two-layer GCN without bias on $\Omega$. If $\Omega$ is class-symmetric about the origin then,*

$$C(L[y]) \leq C[y].$$

*Furthermore, if $\Omega$ is symmetric about $S$ then,*

$$C(P_S[L[y]]) \leq C(L[y])$$

*Proof.* If $\Omega$ is class-symmetric about the origin then,

$$C[y] \geq \mathbb{E}_{x \sim \Omega}\left[\log\left(1 + e^{-v(x)\frac{y(x)-y(-x)}{2}}\right)\right] \tag{33}$$

$$= \mathbb{E}_{x \sim \Omega}\left[\log\left(1 + e^{-v(x)L[y](x)}\right)\right] \tag{34}$$

$$= C[L[y]] \tag{35}$$

Similarly, if $\Omega$ is symmetric about $S$ then the above lemmas applied to $L[y]$ yield,

$$C(L[y]) \geq \mathbb{E}_{x \sim \Omega}\left[\log\left(1 + e^{-v(x)\frac{L[y](x)+L[y](R_S(x))}{2}}\right)\right] \tag{36}$$

$$= \mathbb{E}_{x \sim \Omega}\left[\log\left(1 + e^{-v(x)P_S[L[y]](x)}\right)\right] \tag{37}$$

$$= C(P_S[L[y]]). \tag{38}$$

$\square$

## B.2    ACCURACY ANALYSIS IN THE OPTIMAL-PARAMETER CASE

In light of the preceding theorem, we now study the accuracy of the optimal linear, two-layer GCN, which we are able to compute in integrals. Let

$$y(x) = K \sum_{j \in \mathcal{N}(x)} \sum_{k \in \mathcal{N}(j)} X(k) \cdot m$$

over a cSBM with expected average node degree $d$ and edge information parameter $\lambda$, where $K$ is a constant, direction $m \in \mathbb{R}^{m_{\text{feat}}}$, and features are given by

$$X(i) = v_i \mu m + z_i$$

where $v_i \in \{\pm 1\}$ is the class and $z_i$ is the Gaussian error with mean 0 and variance $\sigma^2 I$. In this case, $X(i) \cdot m$ is given by

$$X(i) \cdot m = v_i \mu + z_i \cdot m = v_i + b_i$$

where $b_i = z_i \cdot m$ is Gaussian with mean 0 and variance $\sigma^2$.

In our analysis, self-loops will be added. Furthermore, $d_{\text{in}}$ and $d_{\text{out}}$ will denote,

$$d_{\text{in}} = \frac{d + \lambda\sqrt{d}}{2}, \quad d_{\text{out}} = \frac{d - \lambda\sqrt{d}}{2}.$$

In the large node limit, the number of neighbors of a node $i$ having the same class, $n_{\text{in}}$, is distributed according to a Poisson distribution with mean $d_{\text{in}}$. Similarly, the number of neighbors having the opposite class, $n_{\text{out}}$, is distributed according to a Poisson distribution with mean $d_{\text{out}}$.

The number of same class neighbors of the same-class neighbors of $i$, denoted $n_{\text{in,in}}$ is given by a Poisson distribution conditional on $n_{\text{in}}$ with mean $d_{\text{in}} n_{\text{in}}$. Similarly the number of opposite class neighbors of the same class neighbors of $i$, denoted $n_{\text{in,out}}$ is given by a Poisson distribution conditional on $n_{\text{in}}$ with mean $d_{\text{out}} n_{\text{in}}$. We define $n_{\text{out,in}}$ and $n_{\text{out,out}}$ similarly.

Let $n_{2-\text{in}}$ and $n_{2-\text{out}}$ denote $n_{\text{in,in}} + n_{\text{out,out}}$ and $n_{\text{in,out}} + n_{\text{out,in}}$ respectively. Intuitively, $n_{2-\text{in}}$ and $n_{2-\text{out}}$ denote the number of same class and opposite class nodes distance two from node $i$. By independence, $n_{2-\text{in}}$ and $n_{2-\text{out}}$ are given by a Poisson distribution conditional on $n_{\text{in}}$ and $n_{\text{out}}$ with means $d_{\text{in}} n_{\text{in}} + d_{\text{out}} n_{\text{out}}$ and $d_{\text{out}} n_{\text{in}} + d_{\text{in}} n_{\text{out}}$, respectively. Then, if we let $p(k, \lambda) = \frac{\lambda^k e^{-\lambda}}{k!}$ be the pmf of the Poisson distribution, the probability of $n_{\text{in}}$, $n_{\text{out}}$, $n_{2-\text{in}}$, and $n_{2-\text{out}}$ occurring can be factored as

$$P(n_{\text{in}}, n_{\text{out}}, n_{2-\text{in}}, n_{2-\text{out}}) \tag{39}$$
$$= p(n_{\text{in}}, d_{\text{in}}) \cdot p(n_{\text{out}}, d_{\text{out}}) \cdot p(n_{2-\text{in}}, d_{\text{in}} n_{\text{in}} + d_{\text{out}} n_{\text{out}}) \cdot p(n_{2-\text{out}}, d_{\text{out}} n_{\text{in}} + d_{\text{in}} n_{\text{out}}). \tag{40}$$

Given $n_{\text{in}}$, $n_{\text{out}}$, $n_{2-\text{in}}$, and $n_{2-\text{out}}$, the model $y(x)$ will have mean

$$\mu K \sum_{j \in \mathcal{N}(x)} \sum_{k \in \mathcal{N}(j)} v_k$$

as the error terms have mean 0. Taking self-loops into account, there are $(n_{\text{in}} + n_{\text{out}} + 1)$ 2-walks to the central node, two 2-walks to each of the neighbors, and one 2-walk to each of the nodes at distance 2. Recall the mean may be calculated linearly while variance satisfies

$$\text{Var}\left(\sum_i a_i X_i\right) = \sum_i a_i^2 \text{Var}(X_i)$$

where the $\{X_i\}_i$ are independent distributions. The conditional mean is therefore given by

$$K\mu v(x)\left((n_{\text{in}} + n_{\text{out}} + 1) + 2(n_{\text{in}} - n_{\text{out}}) + n_{2-\text{in}} - n_{2-\text{out}}\right)$$

and variance

$$K^2\sigma^2\left((n_{\text{in}} + n_{\text{out}} + 1)^2 + 4(n_{\text{in}} + n_{\text{out}}) + (n_{2-\text{in}} + n_{2-\text{out}})\right).$$

When the graph structure is fixed, the model outputs will be Gaussian-distributed (as it is a sum of Gaussian clouds), and its accuracy is the probability that its sign matches $v(x)$. By symmetry, we may assume $v(x) = 1$. If $\Phi$ is the cdf of the standard distribution, then this accuracy is given by $\Phi$ applied to the mean divided by the standard deviation. The accuracy is then,

$$\Phi\left(\frac{K\mu(1 + 3n_{\text{in}} - n_{\text{out}} + n_{2-\text{in}} - n_{2-\text{out}})}{|K|\sigma\sqrt{(n_{\text{in}} + n_{\text{out}} + 1)^2 + 4(n_{\text{in}} + n_{\text{out}}) + (n_{2-\text{in}} + n_{2-\text{out}})}}\right) \tag{41}$$

$$= \Phi\left(\psi\left(\text{sgn}(K)\frac{\mu}{\sigma}, n_{\text{in}}, n_{\text{out}}, n_{2-\text{in}}, n_{2-\text{out}}\right)\right) \tag{42}$$

where

$$\psi(c, n_{\text{in}}, n_{\text{out}}, n_{2-\text{in}}, n_{2-\text{out}}) = c\frac{1 + 3n_{\text{in}} - n_{\text{out}} + n_{2-\text{in}} - n_{2-\text{out}}}{\sqrt{(n_{\text{in}} + n_{\text{out}} + 1)^2 + 4(n_{\text{in}} + n_{\text{out}}) + (n_{2-\text{in}} + n_{2-\text{out}})}}.$$

The total accuracy is then given by

$$\sum_{n_{\text{in}}, n_{\text{out}}, n_{2-\text{in}}, n_{2-\text{out}}=0}^{\infty} P(n_{\text{in}}, n_{\text{out}}, n_{2-\text{in}}, n_{2-\text{out}})\Phi\left(\psi\left(\text{sgn}(K)\frac{\mu}{\sigma}, n_{\text{in}}, n_{\text{out}}, n_{2-\text{in}}, n_{2-\text{out}}\right)\right).$$

# C  COMPLETE SET OF ACCURACY MAPS

## C.1  MEANS

The comprehensive results for mean values of our experiments are found below. Additional experiments using other architectures or wider bounds may be conducted using our code in GitHub

**Binomial Degree Distribution**

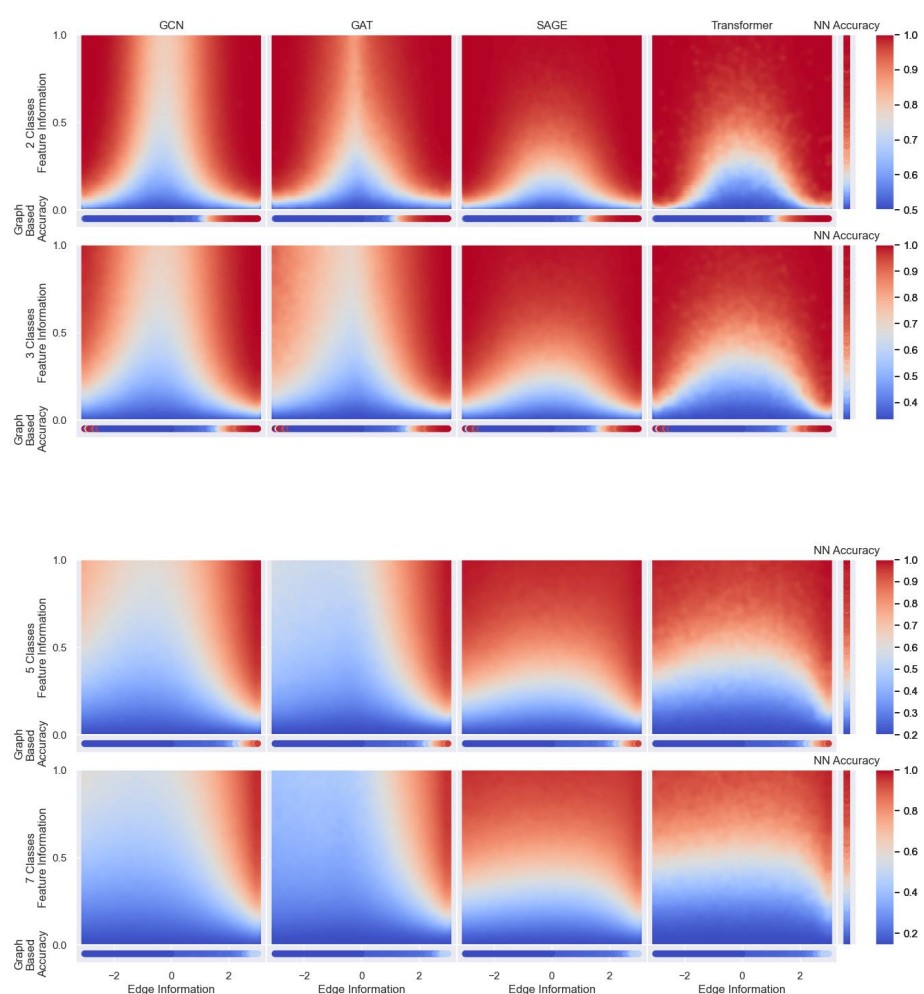

Figure 4: We compare accuracies over the distributed graphs across varying class sizes. We also depict the accuracy curves of a regular feedforward neural network and that of spectral clustering on the same datasets to the right and bottom of each plot respectively. Using these plots, we compare how well each architecture performs on an increased number of classes. Additionally, we view how performance changes across different architectures.

When considering the Binomial SBM, we see that SAGE performed the best of any GNN architecture across any class size. As we increase the number of classes more information is needed for any architecture to classify correctly. Additionally, the increase in class size more adversely affects the heterophilous regime than the homophilous regime. By comparing the figures in fig. 4 and fig. 5 we can observe how each model is affected by degree correction across any number of classes.

In fig. 6 and fig. 7 we view the various regimes across which each architecture outperforms the others. As we increase the class sizes, the favorable regime for the neural network increases in

**Degree-Corrected Degree Distribution**

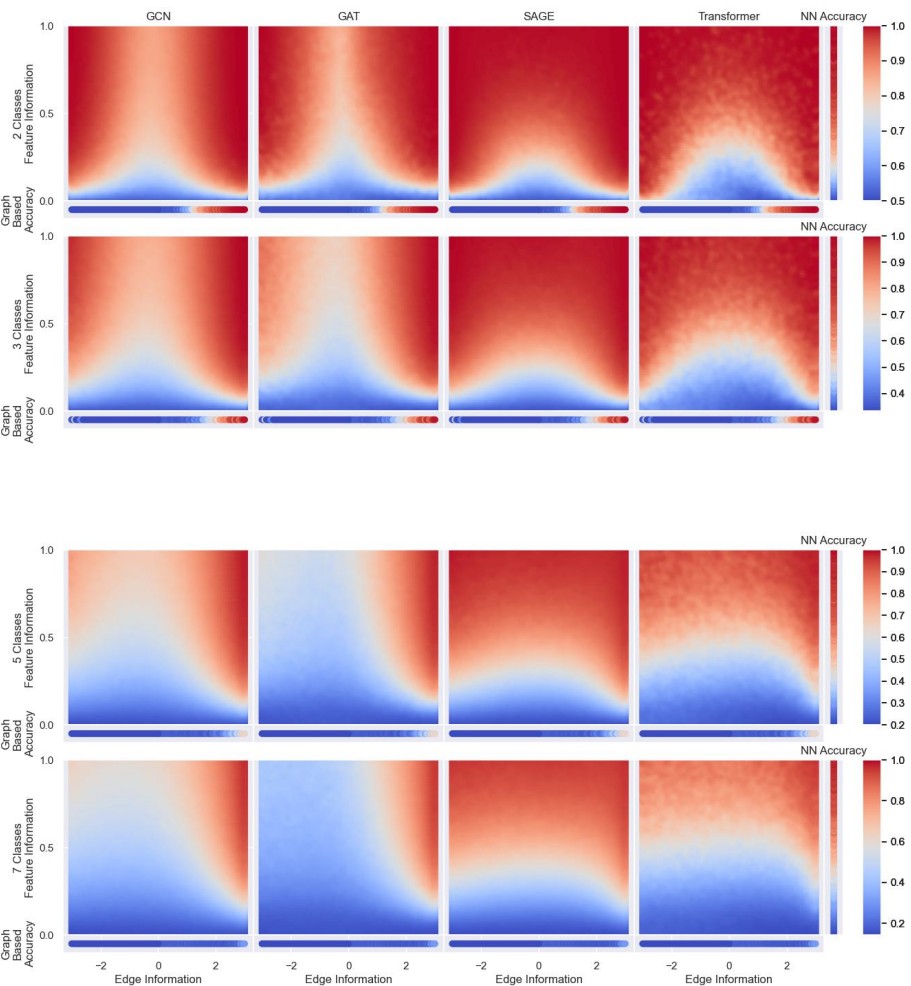

Figure 5: We compare accuracies over degree-corrected graphs with varying class sizes. The GCN did better on degree-corrected graphs across any number of classes. This can be observed by viewing how the blue region in the top figures shrinks in the degree-corrected case. The performance of the Transformer improved in degree-corrected cases for class numbers of two and three, yet it decreased performance for class numbers of five and seven. The performance of SAGE and GAT were mostly unaffected by the degree correction.

size, showing that in many cases it is simply better to ignore edges and utilize solely the feature information. However, it should be noted that in most of the cases, there is always a regime where the GNN architecture outperforms both of the baselines.

**Binomial Degree Distribution**

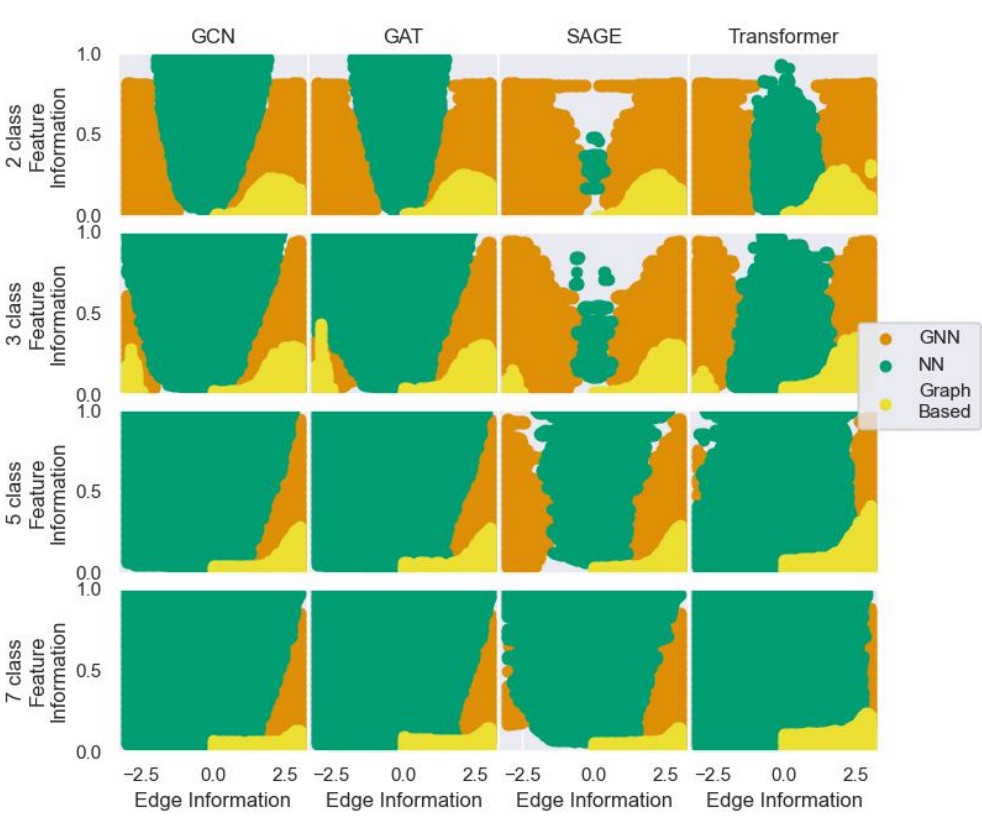

Figure 6: Regions depicting where each architecture outperforms the others across the SBM graphs. Here we can compare how varying parts of the data effects the shape and sizes of the favorable regimes

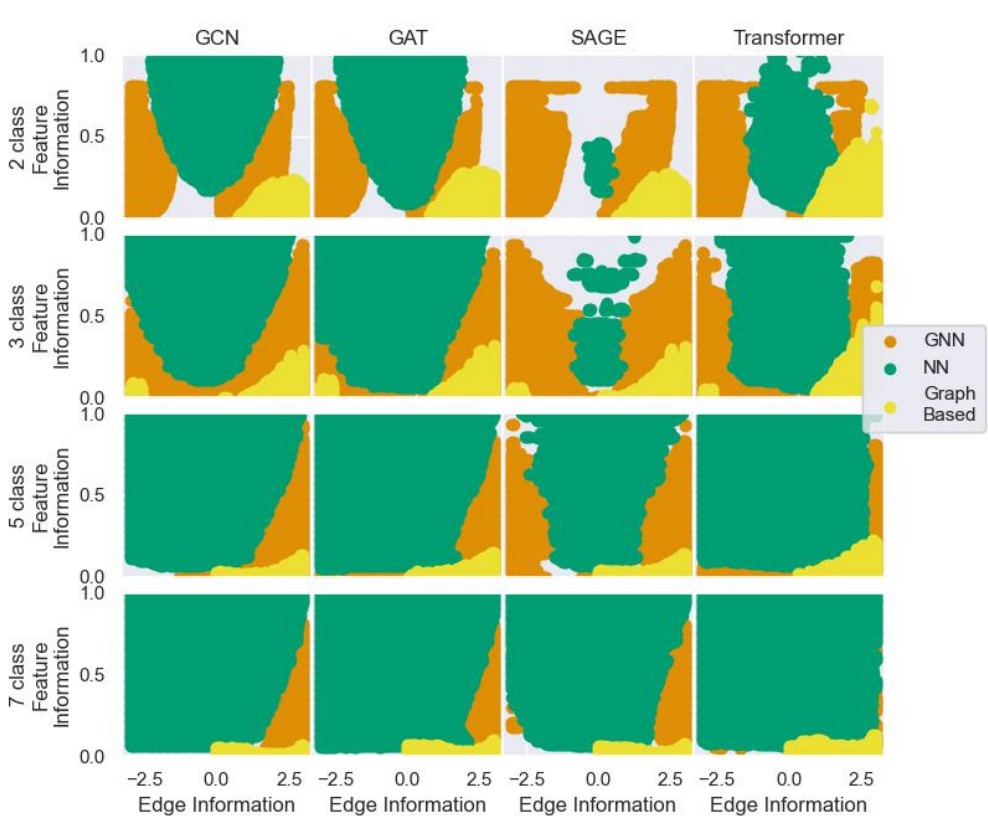

Figure 7: Regions where each architecture outperforms the others across degree-corrected graphs.

## C.2 MAXES

The comprehensive results for max values of our experiments are found below. Additional experiments using other architectures or wider bounds may be conducted using our code in our GitHub.

**Binomial Degree Distribution**

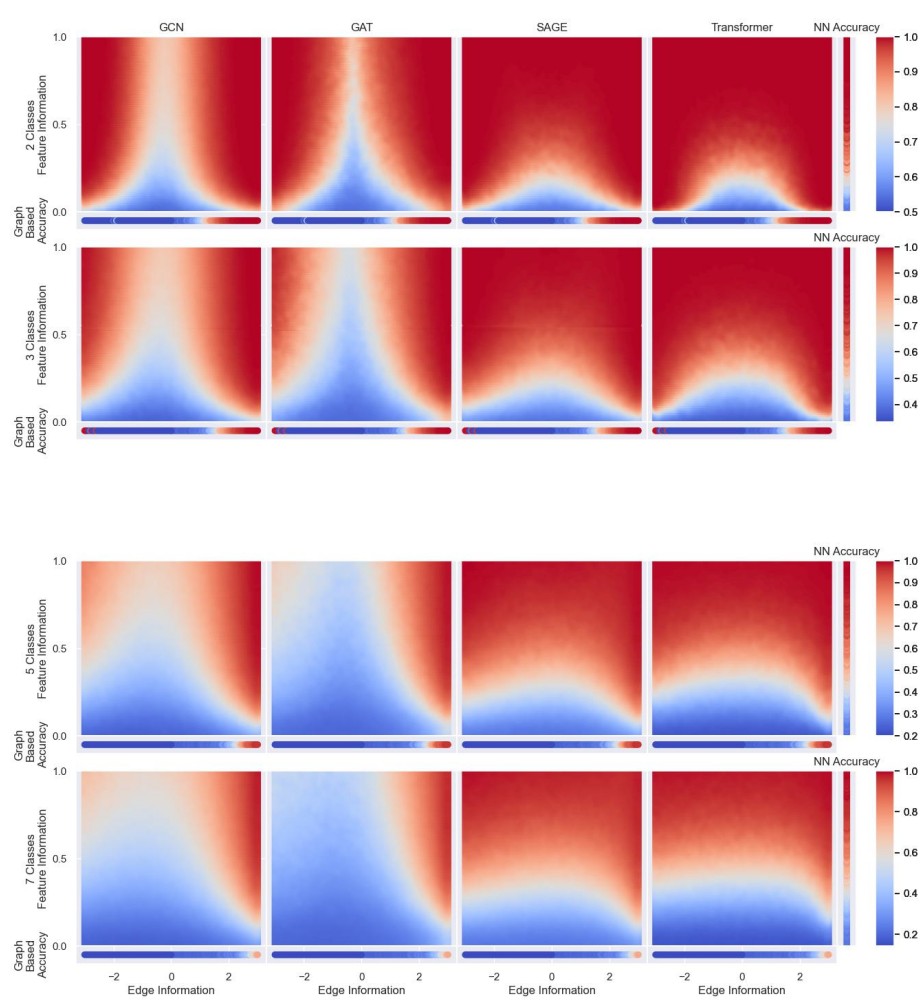

Figure 8: We compare accuracies over the distributed graphs across varying class sizes. Note that the blue regions of these graphs are much more pointed than those of the mean graphs.

NEW

Generally, we find that the Graph-Transformer and SAGE are most resistant to edge and feature noise. In fig. 8 we compare the performance of the GCN, GAT, SAGE, and Transformer on $n$-class, non-degree-corrected cSBM. In these regimes SAGE and Graph are rarely beat by a neural network, showing that they are more resistant to feature noise compared to the GAT or GCN. In addition, SAGE and Transformer achieve perfect classification with zero edge information, something that the GCN and GAT fail to achieve. This reflects that SAGE utilizes more global information (random walks and graph embeddings) than the GCN and GAT, and that the Transformer learns to ignore the graph positional encodings altogether.

NEW

The areas where the models perform the worst (blue areas fig. 8) vary across the architectures, as do the regions where they outperform the neural network and spectral clustering baselines (see fig. 8). For example, the GAT's blue regime goes much higher than the other architectures when there is zero edge information. This suggests that clean features are especially important to the GAT. In

addition, curiously, the degree correction seems to help both the Transformer and the GAT perform slightly better in the heterophilous case (notice the slight skew in blue).

When we view the maxes in light of the average graphs, we see that the blue portions of the maxes are much more steeply shaped than that of the averaged. This likely demonstrates that while both the averages and the maxes perform poorly towards the middle (where there is a lot of edge noise) the model is able to achieve better along the sides of the graph where we have less feature information but more edge information. In general, it seems the models benefited from operating on heavy-tailed

**Degree-Corrected Degree Distribution**

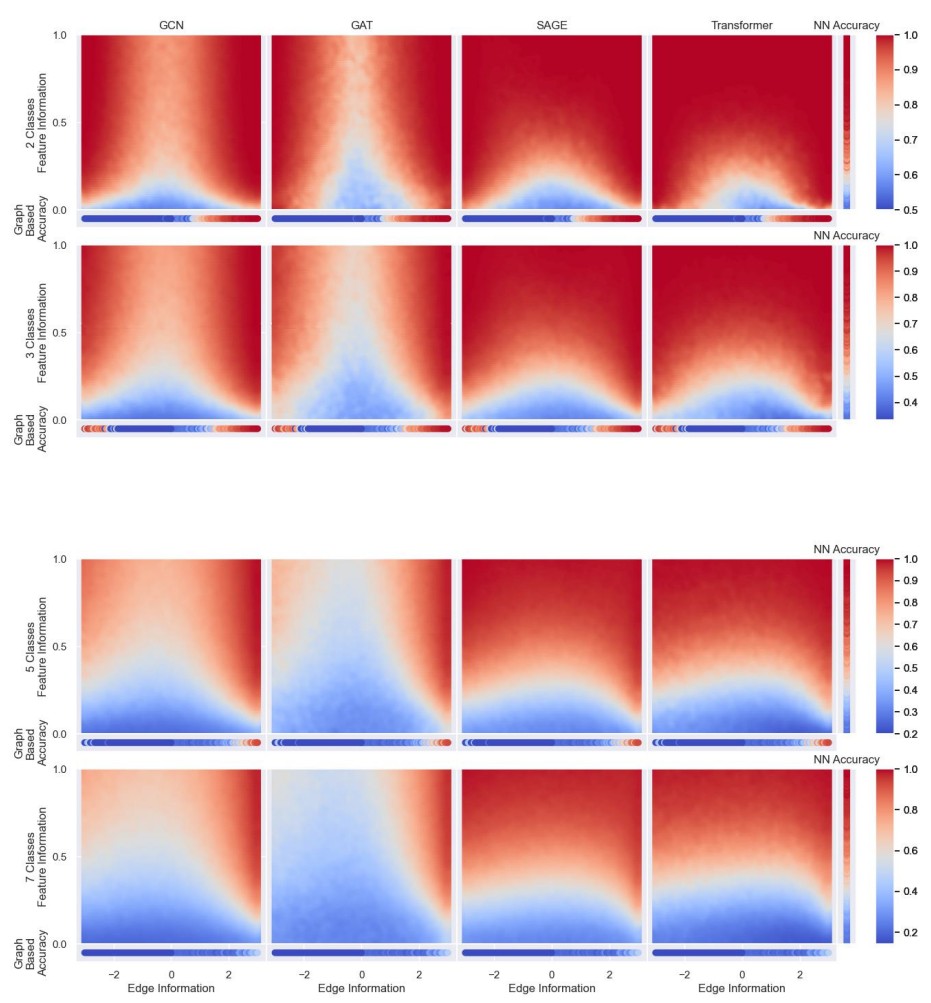

Figure 9: We compare accuracies over degree-corrected graphs across varying class sizes.

graphs. In particular, we see that the GCN and the GAT performed better on degree-corrected graphs across all class sizes. The Transformer and SAGE did not see as stark of an increase in performance, but did perform noticeably better on class sizes of 2 and 3.

## D    FURTHER ANALYSIS OF THE ROLE OF HIGHER-ORDER STRUCTURE

We provide further insights into the effects of higher-order structure in GNNs. To illustrate the impact of structure, we develop several variants of attributed Stochastic Block Models. We track

**Binomial Degree Distribution**

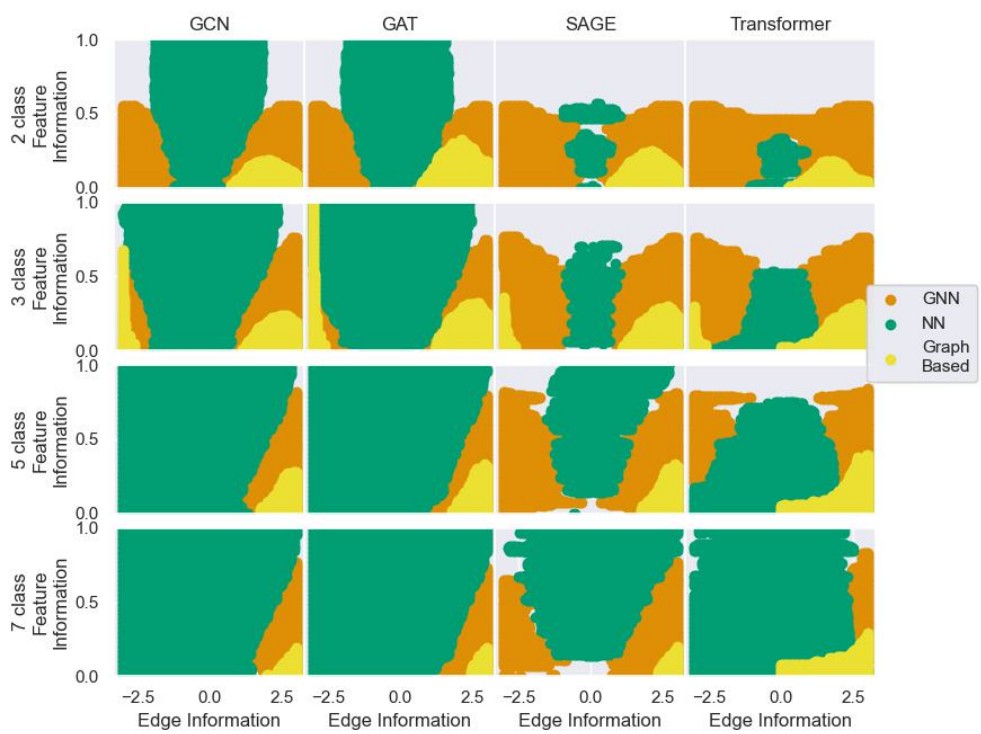

Figure 10: Regions of where each architecture outperforms the others across the SBM graphs. Here we can compare how varying parts of the data effects the shape and sizes of the favorable regimes

**Degree-Corrected Degree Distribution**

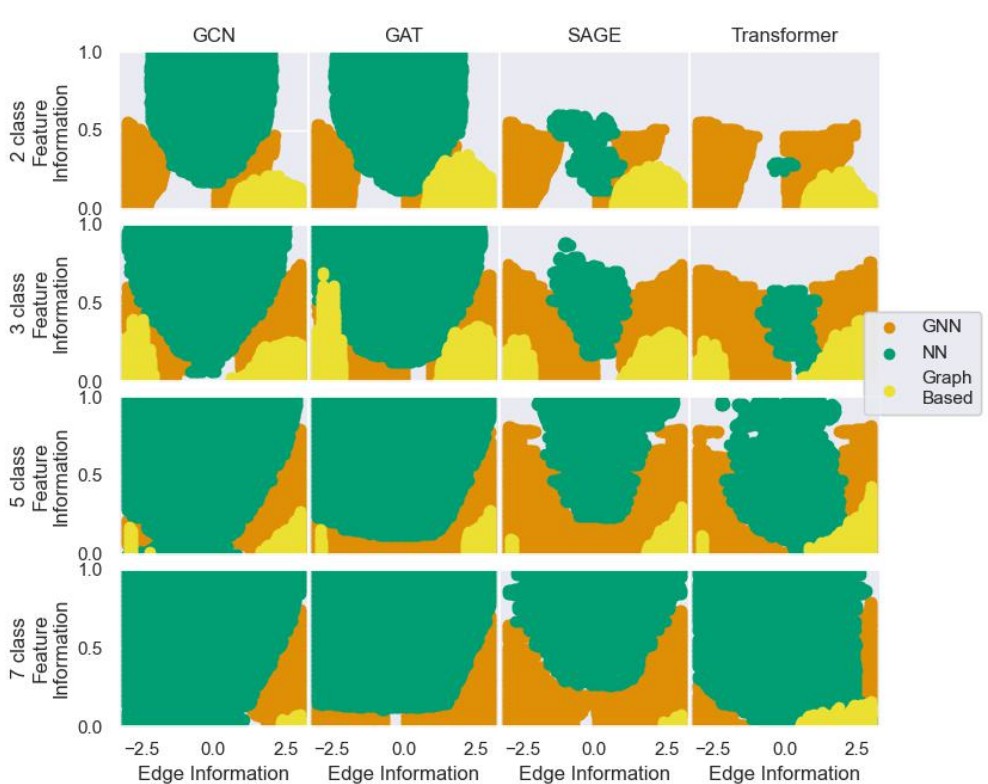

Figure 11: Regions where each architecture outperforms the others across degree-corrected graphs.

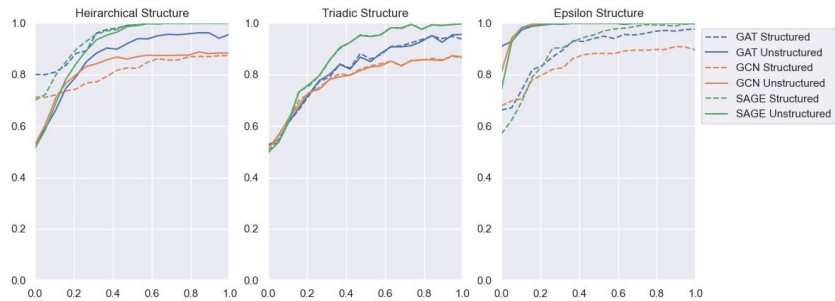

Figure 12: Effects of removing higher-order structure in structured SBMs. We vary feature separability in each of the examples from 0 to 1. GNNs most notably increased under ENN-SBMs.

accuracy on a Hierarchical Stochastic Block Model (hSBM), a Epsilon Nearest Neighbors Stochastic Block Model (ENN-SBM), and a Stochastic Block Model with Triadic Closure.

The implementations can be found in our code on GitHub. To create the hSBM, we generate 5 sub clusters for each class in the SBM that have slightly more similar features and self higher connectivity. We generate an Epsilon Nearest Neighbor Graph by sampling 1000 points from the unit square and randomly assigning half to each class. We then generate the edges by adding an edge between nodes if $||\text{node}_i - \text{node}_j|| \leq \epsilon_{\text{intra}}$ for nodes of the same class for nodes of the same class and $||\text{node}_i - \text{node}_j|| \leq \epsilon_{\text{inter}}$ for nodes of different classes. Lastly, we generate a triadic closed SBM by taking a normal SBM and closing $30\%$ of the possible triadic closures.

We note that, as seen in fig. 12, GNNs perform best on graphs lacking both geographic structure (encoded by ENN-SBM). One possible reason for this is that rewiring the graphs reduces the diameter of a graph, encouraging nodes to be closer to the center of the graph or their own communities. We also note that triadic closure has virtually no effect on the performance of GNNs, while results for hierarchical structure vary across architectures.

Hierarchical structure and spatial structure can both be seen as a version of label noise, as a perfect graph might only connect groups that are relevant to one another. This is not true in all cases, as often both of these attributes can contribute valuable information to a machine learning process.

