# OpenReview forum: "Global minima, recoverability thresholds, and higher-order structure in GNNs"
_ICLR.cc/2024/Conference — Submitted to ICLR 2024_

### Official Review · Reviewer_QsGF · 2023-10-26

**Soundness:** 3 good
**Presentation:** 3 good
**Contribution:** 2 fair
**Rating:** 3
**Confidence:** 3

**Summary:**

This paper studies the performance of Graph Neural Networks (GNN) from three perspective on cSBM datasets.
First, the authors analyze the accuracy performance of GCN on cSBM datasets.
Then, the authors study the empirical performance of four classic GNNs on cSBM datasets with varied properties.
Finally, a discussion of the impact of high-order structure on GNNs is included.

**Strengths:**

1. The accuracy analysis on cSBM datasets reveal the cases when GCNs perform better, and why GCNs achieve optimality on cSBM graphs.
2. The empirical discussion over GNNs' performance on different data regimes is inspiring.
3. The paper is well written and can be clearly understood.

**Weaknesses:**

1. Analysis into heterophily on cSBM datasets have been made in [1][2].
2. The theoretical results are limited to 1-2 layer GCNs, and may not be inspiring enough for designing better GNNs.
3. The empirical discussion is mostly restricted to 2-class cSBM datasets, so a gap exists between empirical results and real-world scenarios.
4. It would be better if more explanation about how the empirical results are affected by the GNN architecture is provided.

[1] Chien, Eli, et al. "Adaptive Universal Generalized PageRank Graph Neural Network." International Conference on Learning Representations. 2020.

[2] Ma, Yao, et al. "Is Homophily a Necessity for Graph Neural Networks?." International Conference on Learning Representations. 2021.

**Questions:**

1. What's the physical meaning about Theorem 3? It would be better if more explanation is included.
2. What's the relationship between theoretical and empirical results?
3. Intuitively, the high-order structure will be affected by the one-hop structure. On a homophilic graph, its one-hop and high-order neighbors would both be homophilic.
How to ensure the high-order impact is completely erased?
Does it suggest that designing high-order GNNs is meaningless?

---

> ### Author Response · Authors · 2023-11-16
> **Response**
>
> Thank you for reviewing, and we are glad you find our empirical results inspiring and that the paper was well written! We have carefully considered your comments and made several major improvements to the paper in response. We believe that the impact of the paper will be much greater because of your feedback. We note that revisions have been highlights in blue for the reviewers convenience.
>
> # Higher Order Structure
> Your comments about higher order structure were particularly interesting to us. First, you asked how we ensured that higher order structure is completely erased, particularly since homophily implies some sort of higher order structure automatically. This is true, and we actually had to clarify some of the language of our paper because this sort of higher order structure is impossible to erase while retaining homophily. Rather, the key point is that other higher order structures can be substantially erased, such as recurring motifs, degree-degree correlations, and spatially correlated edge structures.
>
> The mechanism is a rewiring algorithm where the rewiring ensures that the number of edges between clusters and the degree of each node are respected, but everything else should be randomized. After enough swapping, the resulting graph is basically sampled from a cSBM, which doesn’t have any higher-order structure beyond what is implied by the SBM parameters (e.g. homophily).
>
> As far as we can tell, this does not mean that designing higher-order GNNs is meaningless, in fact maybe just the opposite. Higher order structure is present in data, and low-order GNNs cannot handle it well, but it seems probable that higher order GNNs might be designed that could specifically handle some of these structures (e.g. recurrent motifs) particularly well. This is, of course, the exact sort of question that we hope our work brings attention to.
>
> # Comments on Limitations
> With regards to limitations, we agree that it would be great to handle deep GNNs, but already in the two-layer case, the nonlinearities are unfortunately quite formidable. Fortunately, the 2-layer case is quite commonly used, so we hope that our results are applicable. Similarly, we found the reviewer’s comments about overemphasizing the 2-class empirical results prescient. Our solution has been to change our two figures that used to highlight two-class results to now focus on the more interesting multiclass case. The rest of the discussion section has been rebalanced accordingly to balance binary and multiclass results.
>
> We have also added some additional intuition about Theorem 3 in a Remark. Basically, the accuracy estimate correctly combines the relative abundances of various local graph structures and the degree to which they are helpful.
>
> # Additional Points
> Analysis into heterophily on cSBM datasets have been made in [1][2].
> - We now cite these sources in the appropriate place. In particular, our work agrees with (sometimes contradictory) prior works by showing that performance on heterophilic data sets can be good or bad depending on the strength of the heterophily and the signal strength from the features. We additionally contribute by presenting the accuracy transition in detail and considering a wider range of architectures (e.g. SAGE, Transformer) and cSBM distributions (e.g. degree correction).
>
> It would be better if more explanation about how the empirical results are affected by the GNN architecture is provided.
> - Agreed. We have added some additional discussion to the main text and more substantial additional discussion to Appendix C2.

---

> > ### Comment · Reviewer_QsGF · 2023-11-22
> >
> > Thank you for the authors' comprehensive responses and efforts in revising the manuscript.
> >
> > The updated draft has partially addressed my concerns. Nevertheless, I find that the revised content still falls short in furnishing tangible insights for the enhancement of GNN architectures or in deepening our understanding of specific GNN components. While it is common to incorporate high-order information in current GNN models, the practical applicability of the presented theoretical results remains limited. Consequently, I have decided to keep my original score.

---

### Official Review · Reviewer_yxXE · 2023-11-01

**Soundness:** 2 fair
**Presentation:** 2 fair
**Contribution:** 1 poor
**Rating:** 3
**Confidence:** 3

**Summary:**

This paper investigates the application of graph neural networks to the Contextual Stochastic Block Model (cSBM).

First, the study begins by examining the accuracy of a single node in a single-layer (simple) graph convolutional network without non-linearities, conditioned on the graph structure.  Subsequently, the paper demonstrates that, under certain conditions (which are satisfied by the cSBM), the expected log-likelihood of a two-layer GCN with ReLU activation is lower-bounded by the expected log-likelihood without the ReLU activation. The accuracy for a two-layer (simple) graph convolutional network without non-linearities is also explored, and the formula for the accuracy conditioned on the graph structure is provided, specifically for a single node.

The paper concludes with an extensive set of numerical simulations, benchmarking various graph neural networks on the contextual stochastic block model and on other standard benchmarks.

**Strengths:**

The examination of graph neural networks' performance on contextual stochastic block models is an important and ongoing area of research. The theoretical analysis employs some interesting techniques, including the use of symmetry. Additionally, the experiments are quite extensive and could offer motivation for future work,

**Weaknesses:**

Theoretical results in this paper appear to be somewhat limited, as they focus on computing accuracy at single nodes and rely on fixed GCNs without training. Stronger and more informative findings can already be found in the existing literature, where guarantees for the performance of trained GCNs have been provided (for example, see [1,2])

[1] Wu et al. - A non-asymptotic analysis of oversmoothing in graph neural networks

[2] Baranwal, - Graph convolution for semisupervised classification: Improved linear separability and out-of-distribution generalization

**Questions:**

- Regarding the notation used, it's unclear whether 'W' is meant to represent a row or column vector. If 'z_i' is also a vector, it appears there might be a dimension mismatch in the final equation on page 3.

- Page 5, is the "i" in the equations " ... [X](i)  " a fixed a priori node i ?

---

> ### Author Response · Authors · 2023-11-16
> **Response**
>
> Thank you for your review, and we are glad that you noticed how we employed symmetry in a new way and found our numerical results interesting. We are sure that the paper is stronger with your feedback. We note that revisions have been highlights in blue for the reviewers convenience.
>
> We are somewhat confused with regards to the limitations you see in our theoretical results and would appreciate any clarification you can offer. In short, it appears that you see weaknesses in that (1) we are focusing on the accuracy at single nodes and (2) we do not consider trained GNNs. With respect to the first objection, we perhaps should clarify that, particularly in Theorem 1 (formerly Theorem 2–we renumbered), the accuracy is actually the generalization accuracy over the entire cSBM family, and similarly with Proposition 1, part 3, so we are a bit unclear on what is meant by focusing on single nodes. Secondly, with respect to trained GNNs, our results in (new) Theorem 1 apply to both trained and untrained GNNs, and we even identify the global minimizer and its accuracy. We do not address gradient descent or similar algorithms, but this does not seem to be the spirit of your comment.
>
> Our best guess for the cause of this disconnect is that our paper is fundamentally about non-linear graph models, whereas the papers you referenced in your review focus on linear models. Other reviewers noted various deficiencies in our presentation in that section that obscured our main point. While our paper does reference linear GNNs, the majority of our theory results (and all of our simulations) focus on the behavior of genuinely nonlinear models. We are sometimes able to bound the performance of the nonlinear models in terms of the corresponding linear models. We have substantially rewritten the theory section to more clearly reflect these facts–apologies if the original submission was unclear. Hopefully that section is a much more enjoyable and enlightening read now. All of this is of course a guess, and we welcome clarification if we are off track.
>
> Finally, we note that we have clarified the final section on the detrimental impact of higher order graph structure on GNN performance, clarifying that this section is pointing out a fundamental weakness of current GNN architectures and pointing out possible directions for future improvements.

---

> > ### Comment · Reviewer_yxXE · 2023-11-22
> >
> > Dear Authors,
> >
> > Thank you for taking the time to respond to my comments and for incorporating my suggestions into your manuscript.
> >
> > However, I still believe that both the theoretical and experimental contributions presented in your paper are incremental with respect to the existing literature. For this reason, I will maintain my score.

---

### Official Review · Reviewer_SYz5 · 2023-11-01

**Soundness:** 3 good
**Presentation:** 2 fair
**Contribution:** 2 fair
**Rating:** 3
**Confidence:** 3

**Summary:**

The authors analyze GNN architectures performance on different tasks. Theoretically they establish that for contextual Stochastic Block Model, linear GCN up to 2 layers attains maximum accuracy - in a certain sense. Empirically, they test different GNN architectures on a variety of datasets and study the effect of edge feature and higher-order structures. In conclusion, GNNs are shown to be more suited to learn simpler models such as cSBM and struggles with noisy data and higher-order structure information.

**Strengths:**

Experimental details are interesting and cover a wide range of datasets, both synthetic and from the real world. The conclusion of the paper is thought provoking, in showing that GNN is volatile to higher-order structures, and is more capable of learning simpler, albeit noisy, feature data.

Theoretical results seem correct and self-contained. The take away from the theorems in this paper, in that there are many regimes where linear GNNs are enough to learn the optimal (in some sense) classifier is interesting and corroborated by other research in the literature. The conclusions drawn from the paper is more or less in-tune with current understanding of GNN performance under cSBM, in that nonlinearities need to be re-thought.

**Weaknesses:**

Unfortunately, I believe that the author missed a couple of key literature pieces. For instance, Wu, Chen, Wang and Jababaie ‘A non-asymptotic analysis of oversmoothing in GNN’ in ICLR2023. As a result, some key results of the paper have been shown or can be derived from existing results in the literature. For instance, Theorem 1 of the paper is established in Lemma 1 of Wu, Chen, Wang and Jababaie (granted that this paper do not make clear the distinction between homophily and heterophily regime, the results of Theorem 1 is a few lines away from their Lemma). In Wu et al, the authors also noted the ineffectiveness of nonlinearities (in particular ReLU) in Appendix K1, which may also give rise to much of the results of Theorem 2 in the current paper under review. Therefore, much of the theoretical contribution in this paper has already been established elsewhere.

There are also many (for some, major) notational issues with presentation (a few of which that I caught is deferred to the Question section).

**Questions:**

- Pg 3: n_out “the number of nodes in other classes”, do you mean “the number of neighbors in other classes”? Also, n_in and n_out are very dependent on i - the vertex, so i’d suggest writing n_in(i), for example, to avoid confusion (since you’ve already written \mathcal{N}(i), for example).
- Pg 3: Assuming that the means are of opposite sign does not cause loss of generality only in the case when the number of classes in 2. Can the results be generalized to more than 2 classes?
- Pg 3: First bullet point of Theorem 1. Do you mean “has the same distribution as”? (the equation is a random variable, not a distribution).
- Pg 3: First bullet point of Theorem 1. There seems to be some transpose or dot missing when taking the inner product of vectors (eg - mW should be either m^\top W or m \cdot W).
- Pg 4: Second bullet point of Theorem 1. What is the notation y[X], is it the same as y(X)?
- Pg 4: Third bullet point of Theorem 1. The term “maximum accuracy” also lacks a clear definition in the main paper.
- Pg 6: Statement of Theorem 3. The linear model still has a nonlinear sigma written in it.
- Remark 2: What does “extremely dense” mean? Graphs sampled from SBMs are naturally dense (number of edges of quadratic order of number of vertices).
- Equation 7 (page 14). The first derivative is also 0 when n_in = n_out (is it possible for some setting of the cSBM such that n_in = n_out with high probability?) Perhaps this is mentioned later on below equation 15 but it’s not clear what is the scope of “we make no claims” and if it’s wide enough, should be reflected in the main statement in the main paper. For instance, there are phase transition results for SBM that suggest that if lambda < 1 then the SBM is not distinguishable from Erdos-Renyi model with average degree d.

---

> ### Author Response · Authors · 2023-11-16
> **Response**
>
> # Main items
> Thank you for carefully reading our paper! We found these comments very interesting and are glad that you found the paper thought provoking. We particularly appreciate the pointers to other literature, not all of which we were aware of. We now reference these works in the appropriate places. We do wish to clarify the relationship between our work and these in this comment as well. Revisions have been highlights in blue.
>
> Specifically, in the paper of Wu et al. ICLR 2023, Lemma 1 is indeed closely related to our Theorem 1. There is a difference in normalization of the graph operator and possibly in the homophily assumption, but we expect that these are not essential differences. Indeed, we would not be surprised if versions of our Theorem 1 were known elsewhere. Therefore, we have reclassified our Theorem 1 as a Proposition instead and explicitly cited the related Lemma 1 from Wu et al. nearby to clarify that similar facts were already in the literature.
>
> We do, however, feel that our presentation has caused a lack of clarity about the relationship between our last two theorems and the rest of the literature, including Appendix K1 of Wu et al. For example, Proposition 6 in Wu et al.’s appendix states that adding a ReLU activation function after a sequence of linear aggregation layers does not improve classification. This may have a bearing on our Theorem 1 (now Proposition 1), but we actually view it as quite different from our Theorems 2 and 3 (now Theorems 1 and 2), which deal with a nonlinearity interleaved between the linear layers. More generally, the larger purpose of our theoretical section is to show that the nonlinear models cannot perform better than linear models under certain circumstances. We have rewritten section 4.2 with this distinction in mind and hope that it is clearer for the reviewer.
>
> # Other items:
> Pg 3: n_out “the number of nodes in other classes”, do you mean “the number of neighbors in other classes”? Also, n_in and n_out are very dependent on i - the vertex, so I’d suggest writing n_in(i), for example, to avoid confusion (since you’ve already written \mathcal{N}(i), for example).
> - changed to n_in(i)
>
> Pg 3: Assuming that the means are of opposite sign does not cause loss of generality only in the case when the number of classes in 2. Can the results be generalized to more than 2 classes?
> - We have not generalized it here, but we have explored an increased number of classes in the empirical work. Due to the symmetries it is possible the proof would extend.
>
> Pg 3: First bullet point of Theorem 1. Do you mean “has the same distribution as”? (the equation is a random variable, not a distribution).
> - changed to “has the same distribution as”
>
> Pg 3: First bullet point of Theorem 1. There seems to be some transpose or dot missing when taking the inner product of vectors (eg - mW should be either m^\topW or m\cdot W).
> - changed to m\cdot W
>
> Pg 4: Second bullet point of Theorem 1. What is the notation y[X], is it the same as y(X)?
> - Fixed, changed all to y[X]
>
> Pg 4: Third bullet point of Theorem 1. The term “maximum accuracy” also lacks a clear definition in the main paper.
> - Fixed, changed to Maximum expected accuracy for an arbitrary node
>
> Pg 6: Statement of Theorem 3. The linear model still has a nonlinear sigma written in it.
> - Fixed, we have removed the sigma.
>
> Remark 2: What does “extremely dense” mean? Graphs sampled from SBMs are naturally dense (number of edges of quadratic order of number of vertices).
> - This is an interesting point. Traditionally, in the SBM literature, two main regimes are studied, one where the expected degrees are held constant as the number of nodes approaches infinity, and another where they grow by a logarithmic factor. (For a good review, see Abbe, Community Detection and Stochastic Block Models: Recent Developments, Journal of Machine Learning Research 18 (2018) 1-86). In either case, the graph is far from having O(n^2) edges. Our remark about “extremely dense” refers only to the case where almost all edges are present. We added a clarifying remark.
>
> Equation 7 (page 14). The first derivative is also 0 when n_in=n_out (is it possible for some setting of the cSBM such that n_in = n_out with high probability?) Perhaps this is mentioned later on below equation 15 but it’s not clear what is the scope of “we make no claims” and if it’s wide enough, should be reflected in the main statement in the main paper. For instance, there are phase transition results for SBM that suggest that if lambda < 1 then the SBM is not distinguishable from Erdos-Renyi model with average degree d.
> - Thank you for pointing out this discrepancy in the paper. We would like to clarify this both here and in the appendix. If n_in=n_out then regardless of the weight matrix or theta we have that the probability evaluates to .5 (This can be seen in part 2 of proposition 1). Thus every weight matrix is optimal. We have updated the appendix to reflect this fact.

---

> > ### Comment · Reviewer_SYz5 · 2023-11-21
> >
> > I'd like to thank the the authors for their comments and revision of their draft. I was not able to go into the revised draft into as much details as the first draft but have confirmed that the authors have fixed most of the minor concerns raised. However, I believe that the theoretical contributions are now rather minor. At the same time, I believe that the conclusions drawn from the paper, both theoretical and empirical, is based on the oversimplified model of cSBM - a model class that was originally developed to understand complicated theoretical properties in a sandboxed manner. On these simple datasets, it is not surprising that more complicated structures/architectures are not necessary, but the same conclusion is far from being conclusive on realistic datasets, at least based on the evidence presented in the paper. Therefore I am keeping the same score.

---

### Official Review · Reviewer_WetA · 2023-11-02

**Soundness:** 3 good
**Presentation:** 2 fair
**Contribution:** 2 fair
**Rating:** 5
**Confidence:** 4

**Summary:**

In this paper, the authors study the accuracy of GNNs with respect to the contextual stochastic block model (cSBM) for data generation. In Theorem 1, for one-layer GNNs, they prove some formulae for the accuracy under this model, which are derived based on the Gaussian error function, and then they show that a linear classifier can be as good as the best GNN. In Theorem 3, they derive accuracy formulae for two-layer linear networks. The paper is concluded with experiments.

**Strengths:**

- highly motivated problem
- having many experiments

**Weaknesses:**

- the paper is not well-written; extensive revision is required to make the contributions clear
- the setup is limited and it is not clear whether the linear classifier is good beyond the assumptions
- nice theoretical results but I guess they are highly tied to the assumptions (limited)

**Questions:**

- I think the theoretical contributions of the paper are nice but unfortunately not enough and limited to particular assumptions

- Section 4.2 is not well written. It is not clear what the authors want to say and it does not have flow. I strongly recommend rewriting it.

- The definition of $\sigma$ in page 4 is missing. It first seems it is a function, but apparently it is a constant?


-----------------------------------
After the rebuttal: I appreciate the authors for their response and revision; they have made significant changes to enhance the quality of the paper. Since they addressed my questions and comments, I have decided to slightly increase my score.

---

> ### Author Response · Authors · 2023-11-16
> **Response**
>
> Thank you for taking the time to read our paper! It appears there are two main questions affecting this review: the role of assumptions of linearity and difficulties with the presentation of Theorem 2 (now Theorem 1). We note that revisions have been highlights in blue for the reviewers convenience.
>
> First, regarding Section 4.2, we apologize the section was not easier to read, and we have done a significant rewrite, adding in particular guiding Remarks in appropriate places and clarifying the notation. This theorem is really very nice, along with its proof, and we hope the reviewer will take a second look at the much better presentation. In particular, Theorem 2 implies that Theorem 3 is actually a bound on the accuracy of nonlinear GNNs as well. Your comment has substantially increased the quality of this paper, for which we thank you.
>
> Regarding limitations of GNNs when the assumptions do not hold, this is of course a valid point, and we have rewritten the paper to emphasize that the superiority of linear classifiers in general is not one of the claims of the paper. The object of interest here is the nonlinear GNN, and we are using the performance of linear GNNs as a tool to get there. For example, Theorems 2 and 3 both tell us significant facts about nonlinear GNNs. Namely, unless certain asymmetries are included in the data generation model, there is no theoretical reason that the nonlinearities will be helpful. From this we can get tight GNN accuracy bounds purely in terms of linear models (which turn out to be a special case of the nonlinear GNNs, with the right parameter choices). This motivates research into better models than cSBM for the data and, more importantly, a deeper understanding about what significant features of the data should be incorporated into such a model. For example, ideas like homophily and Gaussian features are simply not enough to correctly motivate nonlinear GNNs theoretically. We have significantly enhanced the presentation and discussion in the paper to help get these points across better.
>
> Smaller point: $\sigma$ was defined on page 4, but its usage here without a subscript is apparently unclear. We have added text explicitly clarifying the role of sigma here.

---

### Meta-Review · Area_Chair_oUB3 · 2023-12-09

**Metareview:**

The paper studies the performance of graph convolutional networks (GCNs) on a data distribution modeled by a contextual stochastic block model (cSBM). The main contributions of the paper are roughly: (1) To show non-linear GCNs are no more helpful than linear GCNs. (2) To derive a formula for the performance of linear GCN. (3) To perform simulations of several popular GNN architectures on the cSBM model in various regimes.

The paper is clear and easy to follow. However, I am unfortunately largely in agreement with SYz5. The cSBM model has been studied in the context of GCN performance, e.g. recently in Wu, Chen, Wang and Jababaie "A non-asymptotic analysis of oversmoothing in GNN". While not identical, the calculations/techniques are fairly reminiscent. Additionally, some of the results (e.g. Theorem 1) heavily rely on symmetries of the data distribution (i.e. are very specific to cSBM), so the applicability of the results beyond cSBM is unclear at best.

**Justification For Why Not Higher Score:**

The techniques for the results are fairly close to prior analyses of the cSBM model in recent works. Moreover, some of the results strongly depend on symmetries of the data model (cSBM), and it's unclear what conclusions can be drawn beyond it.

**Justification For Why Not Lower Score:**

N/A

---

### Decision · Program_Chairs · 2024-01-16

Reject